# A *Drosophila* model of Pontocerebellar Hypoplasia reveals a critical role for the RNA exosome in neurons

Derrick J. Morton[1]*, Binta Jalloh[2,3], Lily Kim[1], Isaac Kremsky[1], Rishi J. Nair[1], Khuong B. Nguyen[1], J. Christopher Rounds[2,3], Maria C. Sterrett[1,4], Brianna Brown[3], Thalia Le[1], Maya C. Karkare[1], Kathryn D. McGaughey[1], Shaoyi Sheng[1], Sara W. Leung[1], Milo B. Fasken[1], Kenneth H. Moberg[3], Anita H. Corbett[1]*

**1** Department of Biology, RRC 1021, Emory University, NE, Atlanta, Georgia, United States of America, **2** Genetics and Molecular Biology Graduate Program, Emory University, NE, Atlanta, Georgia, United States of America, **3** Department of Cell Biology, Emory University School of Medicine, Atlanta, Georgia, United States of America, **4** Biochemistry, Cell and Developmental Biology Graduate Program, Emory University, NE, Atlanta, Georgia, United States of America

* djmorto@emory.edu (DJM); acorbe2@emory.edu (AHC)

**Data Availability Statement:** All relevant data are within the manuscript and its Supporting Information files. The data from the RNA-Seq analysis has also been uploaded at NCBI GEO

## Abstract

The RNA exosome is an evolutionarily-conserved ribonuclease complex critically important for precise processing and/or complete degradation of a variety of cellular RNAs. The recent discovery that mutations in genes encoding structural RNA exosome subunits cause tissue-specific diseases makes defining the role of this complex within specific tissues critically important. Mutations in the RNA exosome component 3 (*EXOSC3*) gene cause Pontocerebellar Hypoplasia Type 1b (PCH1b), an autosomal recessive neurologic disorder. The majority of disease-linked mutations are missense mutations that alter evolutionarily-conserved regions of EXOSC3. The tissue-specific defects caused by these amino acid changes in EXOSC3 are challenging to understand based on current models of RNA exosome function with only limited analysis of the complex in any multicellular model *in vivo*. The goal of this study is to provide insight into how mutations in *EXOSC3* impact the function of the RNA exosome. To assess the tissue-specific roles and requirements for the *Drosophila* ortholog of EXOSC3 termed Rrp40, we utilized tissue-specific RNAi drivers. Depletion of Rrp40 in different tissues reveals a general requirement for Rrp40 in the development of many tissues including the brain, but also highlight an age-dependent requirement for Rrp40 in neurons. To assess the functional consequences of the specific amino acid substitutions in EXOSC3 that cause PCH1b, we used CRISPR/Cas9 gene editing technology to generate flies that model this RNA exosome-linked disease. These flies show reduced viability; however, the surviving animals exhibit a spectrum of behavioral and morphological phenotypes. RNA-seq analysis of these *Drosophila Rrp40* mutants reveals increases in the steady-state levels of specific mRNAs and ncRNAs, some of which are central to neuronal function. In particular, *Arc1* mRNA, which encodes a key regulator of synaptic plasticity, is increased in the *Drosophila Rrp40* mutants. Taken together, this study defines a requirement for the RNA exosome in specific tissues/cell types and provides insight into how

(accession no. GSE147032;https://www.ncbi.nlm.nih.gov/geo/query/acc.cgi?acc=GSE147032).

**Funding:** This work was supported by both a National Institutes of Health F32 grant (GM125350) and a Postdoctoral Enrichment Award from the Burroughs Wellcome Fund to D.J.M, National Institutes of Health F31 grants to B.J. (NS103595) and J.C.R (HD088043), a National Institutes of Health R01 grant (MH107305) to A.H.C. and K.H. M, and a National Institutes of Health R01 grant (GM130147) to A.H.C. D.J.M. was also supported by the Emory University National Institutes of Health Institutional Research and Academic Career Development Award (IRACDA) (GM000680) Fellowships in Research and Science Teaching (FIRST) Postdoctoral Fellowship. The funders had no role in study design, data collection and analysis, decision to publish, or preparation of the manuscript.

**Competing interests:** The authors have declared that no competing interests exist.

defects in RNA exosome function caused by specific amino acid substitutions that occur in PCH1b can contribute to neuronal dysfunction.

## Author summary

Pontocerebellar Hypoplasia Type 1b (PCH1b) is a devastating genetic neurological disorder that preferentially affects specific regions of the brain. Typically, children born with PCH1b have structural defects in regions of the brain including those associated with key autonomic functions. Currently, there is no cure or treatment for the disease. PCH1b is caused by mutations in the RNA exosome component 3 (*EXOSC3*) gene, which encodes a structural component of the ubiquitous and essential multi-subunit RNA exosome complex. The RNA exosome is critical for both precise processing and turnover of multiple classes of RNAs. To elucidate the functional consequences of amino acid changes in EXOSC3 that cause PCH1b, we exploited well-established genetic approaches in *Drosophila melanogaster* that model *EXOSC3* mutations found in individuals with PCH1b. Using this system, we find that the *Drosophila* EXOSC3 homolog (termed Rrp40) is essential for normal development and has an important function in neurons. Furthermore, PCH1b missense mutations modeled in *Rrp40* cause reduced viability and produce neuronal-specific phenotypes that correlate with altered levels of target RNAs that encode factors with key roles in neurons. These results provide a basis for understanding how amino acid changes that occur in the RNA exosome contribute to neuronal dysfunction and disease.

## Introduction

Precise production of both coding and non-coding RNAs requires regulation not only at the level of transcription but also at the post-transcriptional level to ensure precise processing and regulated turnover [1–3]. The significance of post-transcriptional control of gene expression is underscored by the number of diseases associated with defects in RNA processing, surveillance and/or decay machinery [4]. Specifically, mutations in genes encoding structural subunits of the ubiquitously expressed RNA exosome, a critical mediator of both RNA processing and decay, cause a variety of tissue-specific diseases [3, 5, 6].

The evolutionarily conserved RNA exosome complex, first identified in *Saccharomyces cerevisiae* [1, 2, 7], plays critical roles in RNA processing, decay and surveillance. The multi-subunit structure of the RNA exosome complex, which is conserved from archaea to humans, consists of ten subunits [1, 2, 8, 9]. In humans, the RNA exosome subunits are termed EXO-Some Component (EXOSC) proteins. In *S. cerevisiae* and *Drosophila*, most RNA exosome subunits are termed Rrp proteins after the original yeast genetic screen for ribosomal RNA processing (*rrp*) mutants [7]. As illustrated in the **Fig 1A**, the complex is comprised of a catalytically inert core consisting of a hexameric 'ring' of catalytically inactive PH-like ribonuclease domain-containing core subunits 'capped' by a trimer of S1/KH putative RNA binding domain-containing cap subunits and a catalytically active 3' to 5' processive exo- and endoribonuclease, DIS3, which associates with the bottom of the 'ring-like' core. In *S. cerevisiae*, all core RNA exosome genes are essential [1, 2, 10]. Furthermore, in *Drosophila* all RNA exosome genes studied thus far (*Dis3, Mtr3, Rrp6, Rrp41, Rrp42)* are essential and required for normal fly development [11–13]. Studies in mice have been limited to assessing the role of *Exosc3* in B-cells *ex vivo* [14].

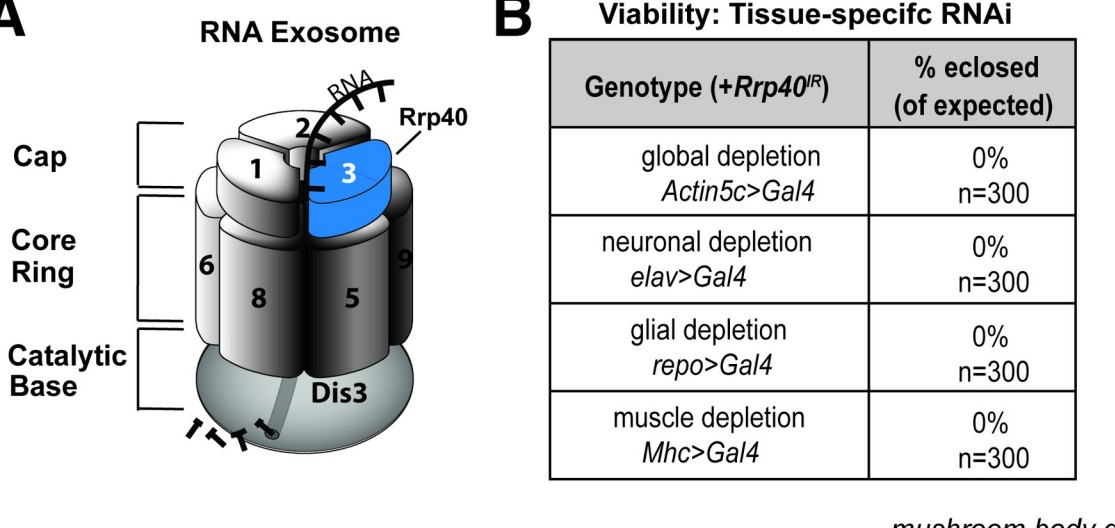

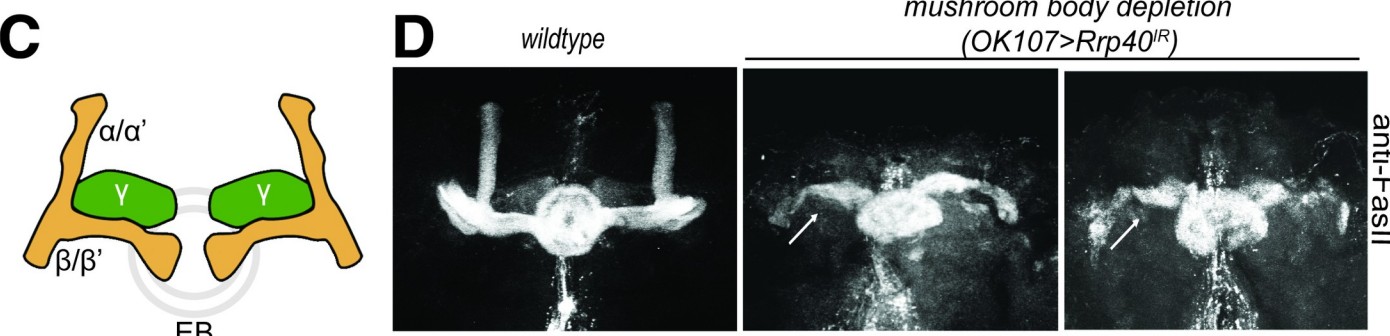

**Fig 1. The Rrp40 subunit of the RNA exosome is required for viability and proper mushroom body development in *Drosophila*.** (A) The RNA exosome is an evolutionarily conserved ribonuclease complex composed of nine structural subunits termed exosome components (EXOSC1-9)—three Cap subunits: EXOSC1; EXOSC2; EXOSC3, six Core Ring subunits: EXOSC4 (not visible); EXOSC5; EXOSC6; EXOSC7 (not visible); EXOSC8; EXOSC9, and a Catalytic Base subunit: Dis3. Mutations in the *EXOSC3* gene encoding a cap subunit [light blue, labeled with 3 (termed Rrp40 in *Drosophila*)] cause PCH1b [6]. (B) Tissue-specific drivers as indicated were employed to knockdown Rrp40 in *Drosophila*. The percentage of flies eclosed (of expected) for the indicated genotypes is shown for a total of 300 flies for each genotype. (C) Diagram of the adult *Drosophila* mushroom body (MB) lobes depicting the vertical alpha (α) and alpha prime (α‘) neurons, the medially projecting (β) and beta-prime (β‘) neurons, and the γ neurons (green) as well as the ellipsoid body (EB) (gray ring). (D) Fasciclin II antibody (anti-FasII) was used to stain either control brains or brains with Rrp40 depleted from mushroom bodies (*OK107>Gal4*). Maximum intensity Z-stack projections of mushroom bodies are shown. β-lobes of control brains do not cross the mid-line and these brains have well-formed α-lobes. Rrp40-depleted mushroom bodies (*OK107-Gal4>Rrp40^{IR}*) (n = 30) have β-lobes defects (white arrows) or have missing α- lobes.

Given that the RNA exosome is essential for viability in model organisms examined thus far, it was surprising when mutations in genes encoding RNA exosome subunits were identified as the cause of several tissue-specific genetic diseases [15]. Mutations in genes that encode structural subunits of the RNA exosome have now been linked to genetic diseases with distinct clinical presentations [reviewed in [16, 17]]. The first such example was the finding that recessive mutations in the *EXOSC3* gene, encoding a structural RNA exosome cap subunit (**Fig 1A**), cause a neurologic disease termed Pontocerebellar Hypoplasia type 1b (PCH1b) [3, 18]. PCH1b is characterized by hypoplasia/atrophy of the cerebellum and pons that leads to severe motor and developmental delays with Purkinje cell abnormalities and degeneration of spinal motor neurons [18–20]. Most individuals with PCH1b do not live beyond childhood. A number of missense mutations that cause single amino acid substitutions in conserved regions of EXOSC3 have been linked to PCH1b [6]. Of note, clinical reports indicate that the severity of PCH1b pathology is influenced by *EXOSC3* allelic heterogeneity, suggesting a genotype-phenotype correlation [6, 21–23]. These observations indicate that EXOSC3 plays a critical role in

the brain and suggest specific functional consequences for amino acid changes linked to disease.

A previous study that modeled loss of *EXOSC3* in zebrafish by knockdown with antisense morpholinos reported embryonic maldevelopment, resulting in small brain size and reduced motility [6]. While co-injection of wildtype zebrafish *exosc3* mRNA could rescue these phenotypes, *exosc3* containing mutations modeling PCH1b failed to rescue [6], suggesting that these mutations disrupt the normal function of EXOSC3. Other studies to assess the functional consequences of RNA exosome disease-linked amino acid changes in RNA exosome subunits have only been performed thus far in yeast [24, 25] providing little insight into how these specific changes could cause tissue-specific consequences. Thus, no studies have produced a genetic model to explore the functional consequences of amino acid changes that cause disease in a multi-cellular model *in vivo*.

In this study, we exploit *Drosophila* as a model to understand the tissue-specific roles and requirements for Rrp40, the *Drosophila* ortholog of EXOSC3. This system also provides an *in vivo* platform to test the functional consequences of amino acid substitutions linked to disease by employing CRISPR/Cas9 gene editing technology to create an allelic series of mutations to model the amino acid substitutions that occur in PCH1b. This work reveals a critical role for Rrp40 in neurons during development and demonstrates that mutations in *Rrp40* in flies, which model genotypes in individuals with PCH1b, cause reduced viability, neuronal-specific behavioral phenotypes, and morphological defects in the brain. RNA-sequencing analysis reveals that *Rrp40* mutants modeling different patient genotypes show differences in levels of mRNAs and ncRNAs that are specific to each mutant providing insight into the molecular basis for the phenotypes observed. For example, we identify the neuronal gene *Arc1*, a critical regulator of neuronal function as increased in the RNA-seq dataset for both *Rrp40* mutants, underscoring the critical role the RNA exosome plays in regulating neuronal transcripts. Taken together, our results provide the first evidence for a critical function of the RNA exosome subunit Rrp40 in neurons. We show that the *Drosophila* model can be used to provide insight into tissue-specific function of the RNA exosome *in vivo* to explore the functional consequences of amino acid substitutions linked to distinct disease phenotypes.

## Results

### The RNA exosome subunit Rrp40 is essential for development

To examine *in vivo* requirements for Rrp40, we employed RNAi to deplete Rrp40 (TRiP *HMJ23923*) from specific tissues/cell types. Successful knockdown of Rrp40 using this RNAi was confirmed by immunoblotting (**S1 Fig**). This RNAi transgene targeting the *Drosophila* EXOSC3 homolog Rrp40 was used in combination with a small panel of four *Gal4* lines with distinct expression patterns: *Actin5c>Gal4* (global); *elav>Gal4* (neurons); *repo>Gal4* (glia); and *Mhc>Gal4* (muscle) to define the requirement for EXOSC3. Expression of each *Gal4* driver to deplete *Rrp40* led to complete lethality, with no surviving adults (**Fig 1B**). This general developmental requirement for *Rrp40* in multiple tissues is consistent with previous studies of other RNA exosome subunits in *Drosophila* [11–13].

### Rrp40 is required for proper development of the *Drosophila* mushroom body neurons

To explore the requirement for Rrp40 in brain development, we utilized one structure in the fly brain that is not essential for viability [26], the mushroom body (MB). As illustrated in **Fig 1C**, each MB contains α/α' and β/β' composed of bundled axons that project from a dorsally

located group of ~2000 Kenyon cells [27, 28]. The β/β' branches project medially toward the ellipsoid and fan-shaped bodies of the midbrain, while the α/α' branches project dorsally [29, 30]. Ellipsoid body neurons normally project axons medially towards the midline and eventually form a closed ring structure [30]. The neuronal adhesion protein Fasciclin-II (FasII), which is enriched on α and β axon branches and regions of the *Drosophila* central complex [30], was used as a marker to assess the effect of loss of Rrp40 on MB structure. This analysis employed a *Gal4* driver specific for MBs (*OK107-Gal4*) in combination with the RNAi transgene targeting *Rrp40* (*UAS-Rrp40^RNAi^; OK107-Gal4*) to deplete Rrp40 specifically from MBs. Optical sectioning of FasII-stained fly brains reveals missing MB α- and β-lobes and outgrowth defects of β-lobes (arrows in **1D**) in the flies with Rrp40 depleted from MBs. We also noted defects in the pattern of FasII staining of the ellipsoid body (EB) in Rrp40-depleted flies. Thus, Rrp40 loss in MBs affects the morphology of the MB α- and β-lobes as well as ellipsoid bodies.

## Rrp40 is required for age-dependent function in neurons

To further investigate the requirement for Rrp40 in neurons, we employed an RNAi temperature shift strategy to bypass the embryonic requirement for Rrp40. As depicted in **Fig 2A**, a ubiquitously expressed, temperature-sensitive *tubulin(tub)-Gal80^ts^* transgene was used to deplete *Rrp40* in combination with the panel of *Gal4* drivers. RNAi was induced by a shift from the permissive to restrictive temperature (18˚C to 29˚C) 24 hours after egg laying. Global *Rrp40* depletion (*Actin5c>Gal4*) using this temperature-shift still yielded no adults, indicating complete lethality; however, *Rrp40* depletion at this time-point in neurons and glia produced some viable adults (**Fig 2B**). In contrast, depletion of *Rrp40* from muscle caused only a mild survival defect. Intriguingly, adult survivors of neuronal *Rrp40* depletion had severely reduced lifespans (~1–2 days) relative to both *Gal4* driver alone and wildtype controls. By contrast, the lifespan of survivors of the glial or muscle depletion is indistinguishable from either control (**Fig 2C**).

To assess the requirement for Rrp40 in neurodevelopment, we exploited this temperature shift strategy to knockdown *Rrp40* expression in neurons in the developing brain (*elav>Rrp40^IR^;tub>Gal80^ts^*) and assess MB development by optical sectioning of FasII-stained brains. As shown in **Fig 2D**, pan-neuronal depletion of Rrp40 (*elav>Rrp40^IR^;tub>Gal80^ts^*) elicits MB defects, including thinned α-lobes and/or fused β-lobes (**Fig 2D**). In sum, these RNAi data argue for a strict requirement for *Rrp40* and thus RNA exosome function in multiple tissues during embryonic development. Bypassing this embryonic requirement reveals an enhanced requirement for Rrp40 in neurons during later stages of development and adult homeostasis, including the branching and projection of axons into the α- and β-lobes.

## PCH1b mutations engineered into the *Rrp40* locus impair viability and shorten lifespan

A number of missense mutations encoding single amino acid substitutions in conserved regions of EXOSC3 have been linked to PCH1b [15]. As shown in **Fig 3A**, The EXOSC3/ Rrp40 protein consists of three domains: an N-terminal domain, an S1 putative RNA binding domain, and a K-homology (KH) putative RNA binding domain. Human EXOSC3 Glycine 31 [G31] and the analogous *Drosophila* G11 residue are located in a conserved region within the N-terminal domain, while EXOSC3 Aspartate 132 [D132] and the analogous *Drosophila* D87 residue are located in a conserved region within the S1 domain (**Fig 3A**).

We utilized CRISPR/Cas9-mediated genome editing technology to create alleles of Rrp40 corresponding to amino acid substitutions in EXOSC3 that cause PCH1b disease (**Fig 3B**). As illustrated in **Fig 3B**, *Drosophila* allows for true modeling of both homozygous and

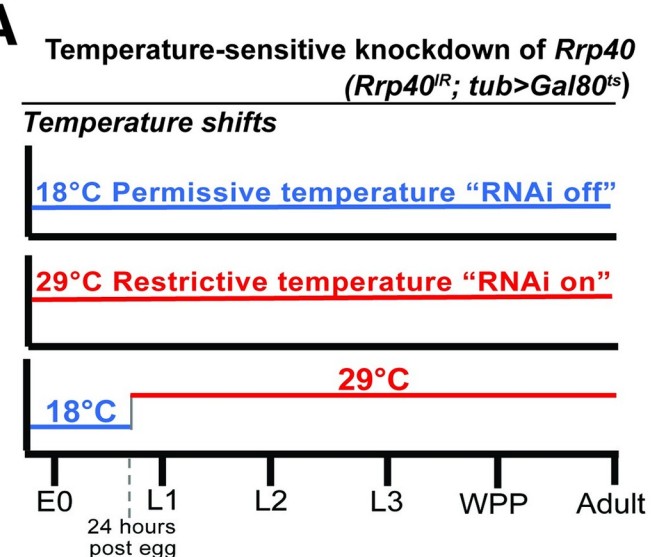

**A**

### Temperature-sensitive knockdown of *Rrp40* (*Rrp40^{IR}; tub>Gal80^{ts}*)

*Temperature shifts*

**18°C Permissive temperature "RNAi off"**

**29°C Restrictive temperature "RNAi on"**

**18°C** → **29°C**

E0 — 24 hours post egg laying — L1 — L2 — L3 — WPP — Adult

**B**

### Frequency of Post-embryonic RNAi Survivors

| Genotype (+*Rrp40^{IR}; tub>Gal80^{ts}*) | % eclosed (of expected) |
|---|---|
| global depletion *Actin5c>Gal4* | 0% n=300 |
| neuronal depletion *elav>Gal4* | 6% n=300 |
| glial depletion *repo>Gal4* | 1% n=300 |
| muscle depletion *Mhc>Gal4* | 63% n=300 |

**C**

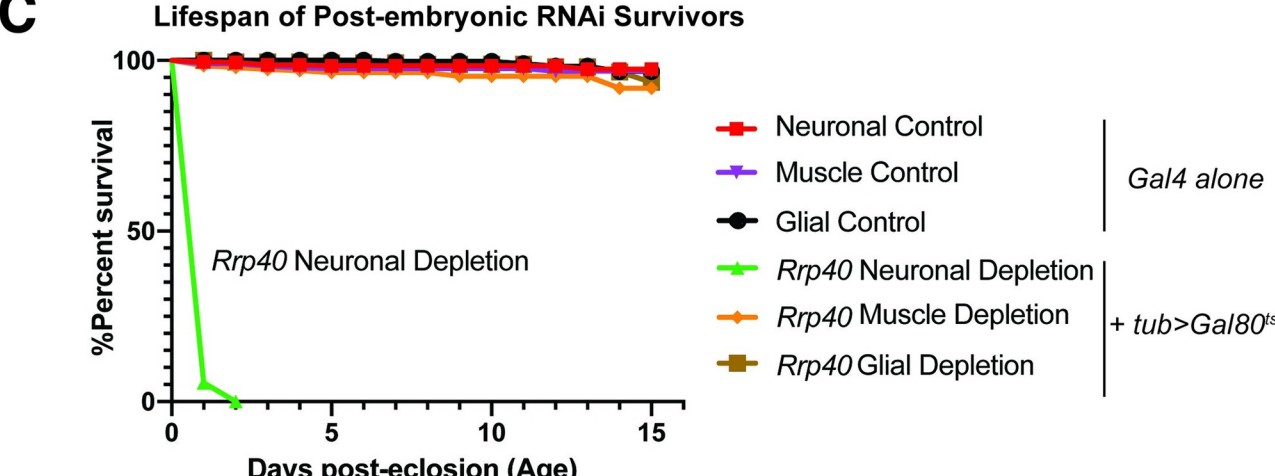

### Lifespan of Post-embryonic RNAi Survivors

Neuronal Control
Muscle Control — *Gal4 alone*
Glial Control

*Rrp40* Neuronal Depletion
*Rrp40* Muscle Depletion — *+ tub>Gal80^{ts}*
*Rrp40* Glial Depletion

*Rrp40* Neuronal Depletion

x-axis: Days post-eclosion (Age)
y-axis: %Percent survival

**D**

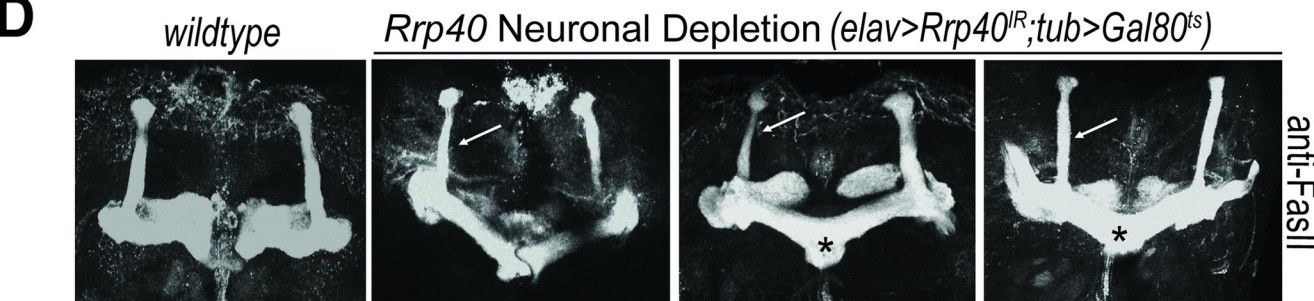

*wildtype* | *Rrp40* Neuronal Depletion (*elav>Rrp40^{IR};tub>Gal80^{ts}*)

anti-FasII

**Fig 2. Rrp40 is required for age-dependent function in neurons.** (A) A scheme to illustrate the approach to achieve temperature-sensitive knockdown of Rrp40 is shown: The *Gal80^{ts}* allele shifts between 18˚C (blue, RNAi off) and 29˚C (red, RNAi on). E0 = 0hr embryo, L1/2/3 = 1st, 2nd or 3rd instar larvae; WPP = white prepupa. (B) The viability of flies in which temperature sensitive RNAi was employed to deplete Rrp40 24 hours post egg laying is shown. The viability of these post-embryonic RNAi survivors as percentage of flies eclosed (of expected) is shown for the indicated tissue-specific drivers. (C) Kaplan-Meier analysis of post-embryonic RNAi survivors for either control (*Gal4* alone) or temperature sensitive knockdown (*tub> Gal80^{ts}*) for the following tissue-specific drivers: Neuronal (*elav>Gal4*, n = 50); Muscle (*Mhc>Gal4*, n = 50); or Glial *(repo>Gal4*, n = 50*)*. (D) Fasciclin II antibody (anti-FasII) staining of both wildtype control brain and three representative brains with temperature-sensitive neuronal depletion of Rrp40 (*elav>Rrp40^{IR};tub>Gal80^{ts}*) are shown. Maximum intensity Z-stack projections of mushroom bodies are shown. β-lobes in control wildtype brains do not cross the mid-line and these brains have well-formed α and β-lobes (n = 30). Mushroom

bodies from flies with Rrp40 depleted from neurons (*elav>Rrp40^IR;tub>Gal80^ts*) (n = 5) have thinned α-lobes (white arrows) and β-lobes that often project to the contralateral hemisphere and appear to fuse (black asterisks).

heterozygous alleles that cause disease. In parallel, a control *Rrp40* chromosome was also created using a wildtype Homology Directed Repair (HDR) donor template to create flies we refer to as *Rrp40^wt* that had undergone the same editing process as flies with engineered base substitutions. As described in Materials and methods and illustrated in **S2 Fig**, all three *Rrp40* alleles (*Rrp40^wt*, *Rrp40^G11A*, *and Rrp40^D87A*) contain the *3xP3-DsRed* cassette inserted in the intergenic region downstream of the *Rrp40 3'UTR*, which allows for visual sorting by red fluorescence in eye tissue. The HDR donor templates containing *3xP3-DsRed* were injected into *nos*-Cas9 embryos. To confirm targeting events in DsRed-positive flies, genomic DNA was amplified and sequenced in the Rrp40 wildtype (*Rrp40^wt*) control flies and in flies containing missense mutations (independent flies, n = 5) (*Rrp40^G11A* and *Rrp40^D87A*) (**S3 Fig**).

Initial analysis of these missense *Rrp40* alleles revealed reduced homozygote viability to adulthood within intercrosses of heterozygous parents (*Rrp40^G11A*: 14% of expected, n = 126, and *Rrp40^D87A*: 38% of expected, n = 182) (**Fig 3B**). In contrast, *Rrp40^wt* control flies eclose at a rate comparable to wildtype. The numbers of surviving adults were further reduced by placing *Rrp40^G11A* or *Rrp40^D87A* over a genomic deletion (*(2L) Exel600*) that removes the *Rrp40* locus, suggesting that both of these missense alleles are hypomorphs, rather than amorphs or hypermorphs (**Fig 3B**). The lower viability of the *Rrp40^G11A* allele, either *in trans* to itself or to (*Df (2L) Exel6005*), indicates that this is a stronger loss-of-function allele than *Rrp40^D87A*.

Significantly, flies with the engineered *Rrp40^wt* allele have a similar lifespan to control *wildtype* (*w^1118*) flies within the tested 30-day testing period, showing that the *3xP3-DsRed* cassette inserted into the intergenic region downstream of *Rrp40* does not appreciably impair survival (**Fig 3C**). Thus, *Rrp40^wt* was included as a control for subsequent studies. In contrast to Rrp40^wt, Kaplan-Meier plots of *Rrp40^G11A* homozygotes and *Rrp40^D87A* homozygotes reveal significantly impaired survival, with all *Rrp40^G11A* animals dead by day 14 and all *Rrp40^D87A* homozygotes dead by day 25 (**Fig 3C**). We are also able to assess the lifespan of flies with the *Rrp40^D87A* allele in trans to a deficiency (*Df(2L) Exel6005*), which mimics a reported patient genotype (**Fig 3C**) [6]. These flies have the shortest lifespan of all genotypes analyzed. As no flies with *Rrp40^G11A* in trans to a deficiency (*Df(2L) Exel6005*) were produced, we could not assess lifespan. Indeed, no patient with EXOSC3 Glycine 31 [G31] over a deletion has been reported (Not detected, ND in **Fig 3B**).

### *Rrp40* mutant flies exhibit age-dependent morphological and behavioral defects

*Rrp40^G11A* and *Rrp40^D87A* adult survivors exhibit wing posture and behavioral defects that are consistent with neurological deficits [31]. *Rrp40^G11A* and *Rrp40^D87A* adults position their wings in an abnormal "held-up and held-out" posture rather than folding them together over the dorsal surface of the thorax and abdomen (**Fig 4A**) and this phenotype worsens with age (**Fig 4B**). *Rrp40^G11A* and *Rrp40^D87A* mutant animals also show reduced locomotor activity in a negative geotaxis assay that becomes more severe with age (**Fig 4C**). Both genotypes also display bouts of spasm interposed with periods of paralysis (**S1, S2 and S3 Videos**) relative to age-matched control *Rrp40^wt* flies. The severity of the wing posture and locomotor defects among *Rrp40* alleles follow a similar pattern: *Rrp40^G11A* causes a more severe phenotype than *Rrp40^D87A*, which is in turn is enhanced by (*Df(2L) Exel6005*). Notably, this pattern of phenotypes mimics the pattern of disease severity observed among individuals with PCH1b [15].

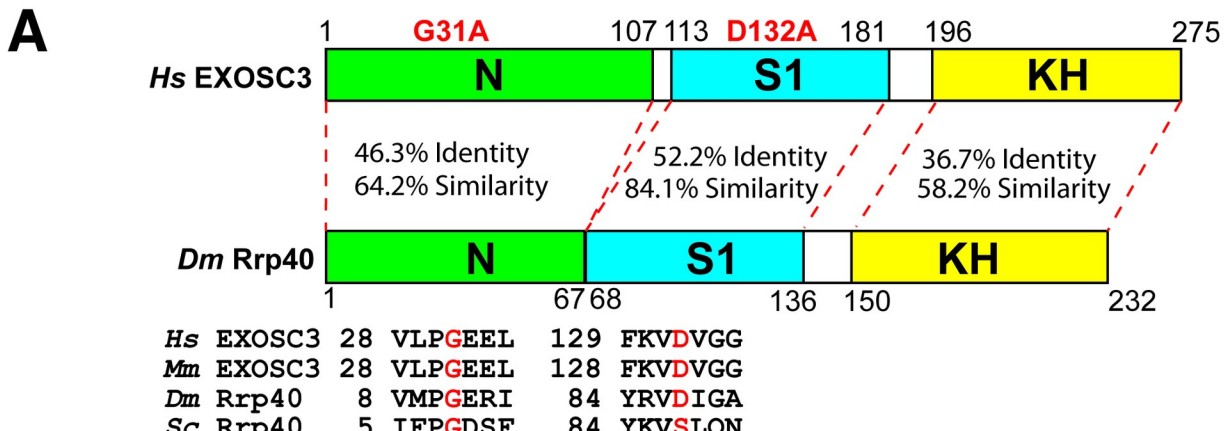

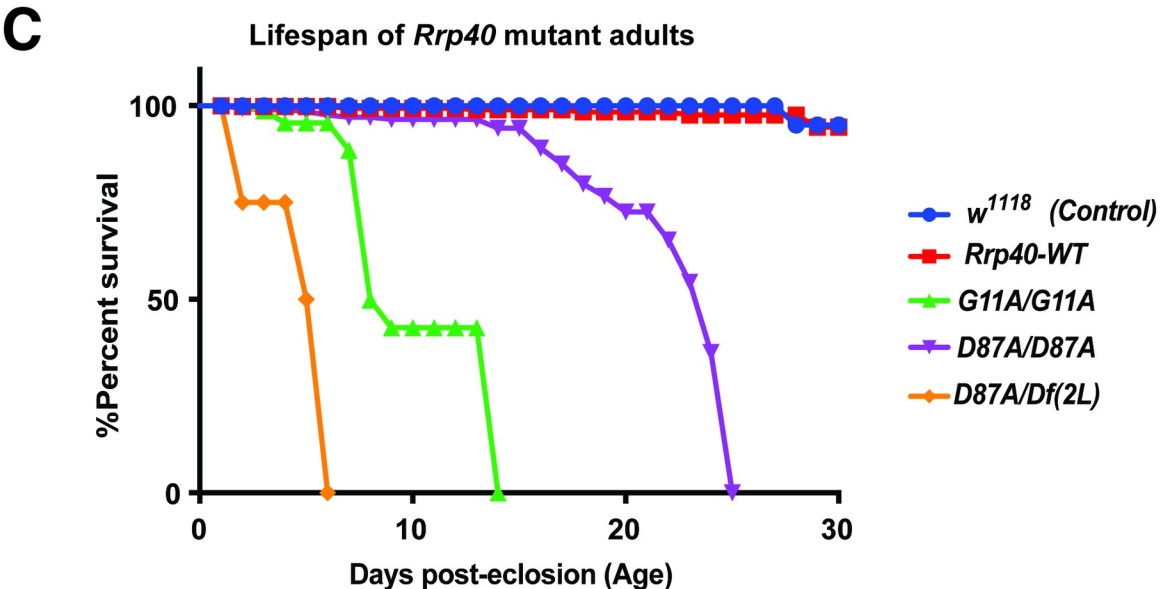

**A**

**B**

| EXOSC3 | Rrp40 | Patient Genotype | *Drosophila* Genotype | % eclosed (of expected) |
|---|---|---|---|---|
| G31A | G11A | G31A Homozygous | G11A Homozygous | 14% n=126 |
| | | ND | G11A/Df(2L) Heterozygous | 0% n=70 |
| D132A | D87A | D132A Homozygous | D87A Homozygous | 38% n=182 |
| | | D132A/null Heterozygous | D87A/Df(2L) Heterozygous | 15% n=160 |

**C**

**Fig 3. Generation of *Drosophila* models of PCH1b amino acid substitutions.** (A) The EXOSC3/Rrp40 protein consists of three domains: an N-terminal domain (N), a central S1 putative RNA binding domain, and a C-terminal putative RNA binding KH (K homology) domain. The disease-causing amino acid substitutions (G31A and D132A) modeled in *Drosophila* in this study are indicated in red above the domain structure. The position and flanking sequence of the amino acid substitutions in PCH1b-associated human EXOSC3 (shown in red) and the corresponding amino acids in *Mus musculus* (*Mm*), *Drosophila melanogaster* (*Dm*), and *Saccharomyces cerevisiae* (*Sc*) are shown. The percent identity/similarity between the human and *Drosophila* domains is indicated. (B) The amino acid substitutions that occur in EXOSC3 in patients that are modeled in *Drosophila* Rrp40, are indicated with Patient Genotype (Homozygous/Heterozygous) as modeled in *Drosophila* (*Drosophila* Genotype) shown. For each *Drosophila* model, the viability of that genotype engineered in flies by CRISPR/Cas9 gene editing is shown as % eclosed (of expected). *Rrp40* mutants show decreased viability, indicated by skewed Mendelian ratios. n = the number of individual flies analyzed. The *G11A* over null genotype has not been reported in patients-indicated as not detected (ND) under patient genotype. (C) A Kaplan-Meier analysis shows that *Rrp40* mutant flies with the indicated genotypes to model patient genotypes exhibit early adult lethality compared to wildtype Control flies (*w1118*) or *Rrp40-WT* (*Rrp40wt*) flies that have undergone CRISPR/Cas9 gene editing to produce a control wildtype *Rrp40* allele (see Materials and methods).

## *Rrp40G11A* mutant flies have a mushroom body β-lobe midline-crossing defect

We analyzed MB structure in wildtype and *Rrp40* mutants by immunostaining for FasII in whole adult brains. Confocal microscopy of *Rrp40wt* flies (n = 18), reveals no apparent MB defects in bifurcation or projection of α- and β-lobes neurons (***Rrp40wt*, Fig 5A**). In contrast, brains of *Rrp40G11A* mutants show β-lobe fibers encroaching toward the midline, 56% (n = 23) (**Fig 5A**), sometimes sufficient to cause apparent fusion of the left and right β-lobes (**Fig 5A**). In contrast, *Rrp40D87A* mutant flies (n = 18) show no detectable defects in α- or β-lobes (***Rrp40D87A*, Fig 5**).

## Pan-neuronal expression of human *EXOSC3* is sufficient to rescue behavioral phenotypes in *Rrp40* mutant flies

To both ensure that phenotypes observed are due to the engineered *Rrp40* mutations and explore functional conservation between Rrp40 and human EXOSC3, we created transgenic flies that express human *EXOSC3-myc* from a UAS inducible promoter. The *UAS-EXOSC3-myc* transgene was expressed in the neurons (*elav>Gal4*) or muscle (*Mhc>Gal4*) of *Rrp40* mutant animals (**Fig 6A**). Notably, *UAS-EXOSC3-myc* produced a mild but consistent rescue of *Rrp40* mutant locomotor defects in the absence of the *Gal4* driver (*rescue control*, **Fig 6B**), suggesting leaky expression from the transgene. Addition of the *elav>Gal4* (neuronal) driver led to a robust rescue of climbing behavior: *Rrp40* mutant animals with neuronal expression of *EXOSC3* (*elav>EXOSC3*) perform normally in locomotor assays (**Fig 6B**); however, *Rrp40* mutant animals with expression of *EXOSC3* in muscle (*Mhc>EXOSC3*) did not show any statistically significant rescue (**Fig 6B**). Thus, human *EXOSC3* can restore functions that are lacking in adult *Rrp40* mutant neurons. This phenotypic rescue also confirms that the locomotor phenotype observed is due to the introduced CRISPR/Cas9 mutations.

## Steady-state levels of RNA exosome subunit levels are affected in *Rrp40* mutant flies

Amino acid substitutions in Rrp40 could alter protein steady-state levels or impact the levels of other RNA exosome complex components. To test this possibility *in vivo*, we compared wildtype and Rrp40 variant levels in fly heads. As shown in **Fig 7A**, we performed immunoblotting of RNA exosome cap subunits (Rrp4 and Rrp40) and a core ring subunit (Mtr3) (**see Fig 1A**). We quantitated the results for Rrp40 and Rrp4, which reveals that levels of these cap subunits are modestly reduced in *Rrp40* mutant fly heads (**Fig 7B**) compared to control. However, the decrease in levels of Rrp40 and Rrp4 is not statistically significantly different between the two

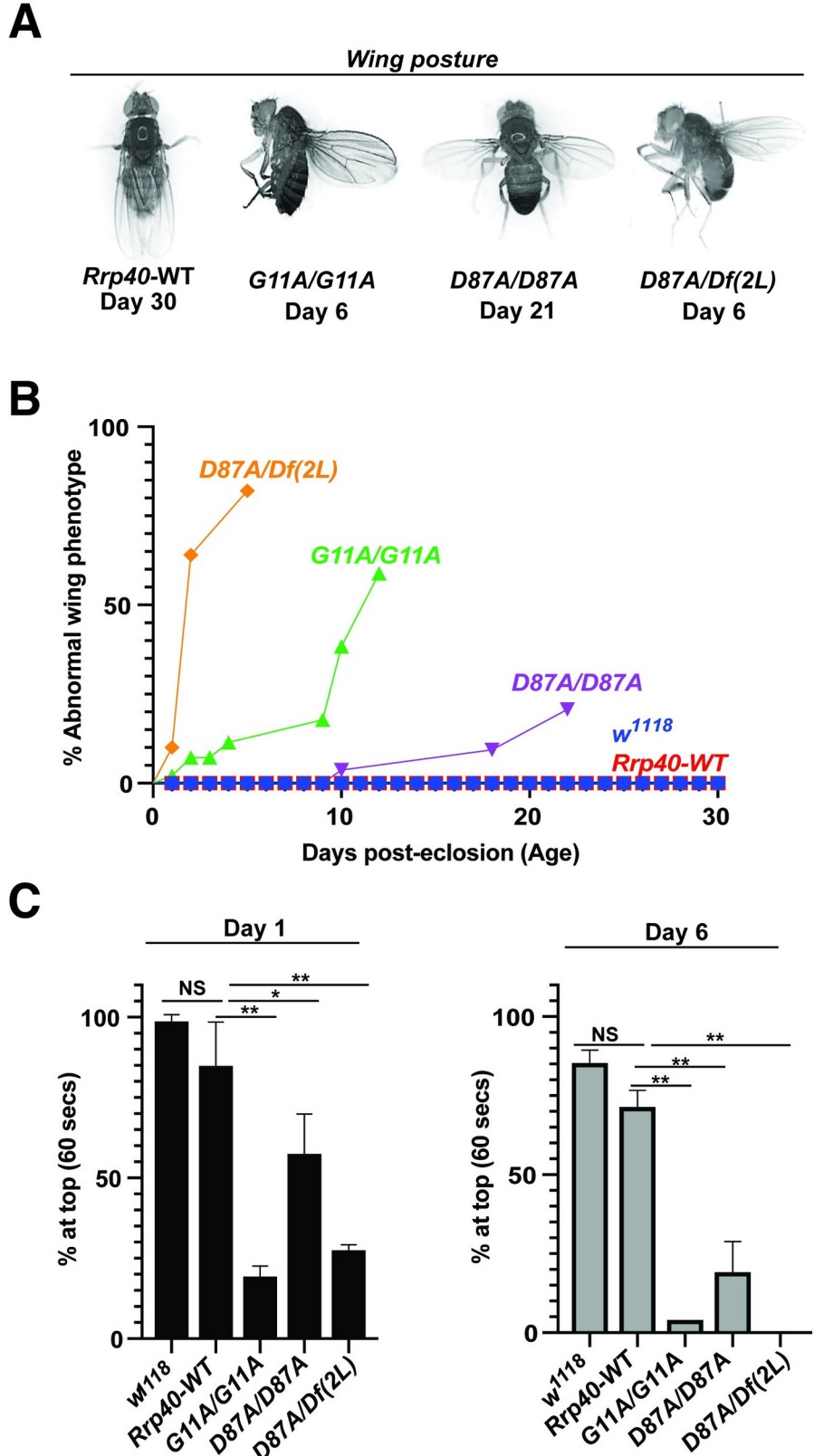

**Fig 4. Flies that model PCH1b amino acid substitutions in *Drosophila* show a range of morphological and behavioral phenotypes.** (A) *Rrp40* mutant flies show a progressive wing position phenotype in adults. Abnormal wing

posture phenotypes (wings held-up or -out) are shown for *Rrp40* mutant flies compared to *Rrp40-WT* flies with Days post-eclosion indicated for each representative image. (B) Quantitative analysis of the abnormal wing position phenotype of *Rrp40* mutants raised at 25˚C. The percentages of adults with abnormal wing position over time (Days post-eclosion) were calculated based on 60 flies per genotype. Neither $w^{1118}$ nor Control *Rrp40-WT* flies showed any abnormal wing position (0% abnormal wing posture, n>100) over the time course analyzed. (C) Locomotor activity was assessed using a negative geotaxis assay as described in Materials and methods. Data are presented as the average percentage of flies of the indicated genotypes that reach the top of a cylinder after 60 seconds across all trials. Groups of 10 age-matched flies [Day (1) and Day (6)] were tested for at least three independent trials per genotype. Values represent the mean ± SEM for n = 3 independent experiments. Asterisks (*) indicate results that are statistically significance at *p-value < 0.05; **p<0.01. Results that show no statistical significance when compared are indicated by NS.

mutants, suggesting these decreases are not sufficient to explain the phenotypic differences detected.

## RNA-sequencing analysis of *Rrp40* alleles reveals distinct gene expression profiles

To identify relevant RNA targets of the RNA exosome, we performed RNA-Seq analysis on three biological replicates of adult heads for each of the following genotypes ($Rrp40^{wt}$, $Rrp40^{G11A}$ and $Rrp40^{D87A}$) from newly eclosed (Day 0) flies. Unbiased principal component analysis (PCA) of the resulting RNA expression data produced three distinct clusters (**Fig 8A**),

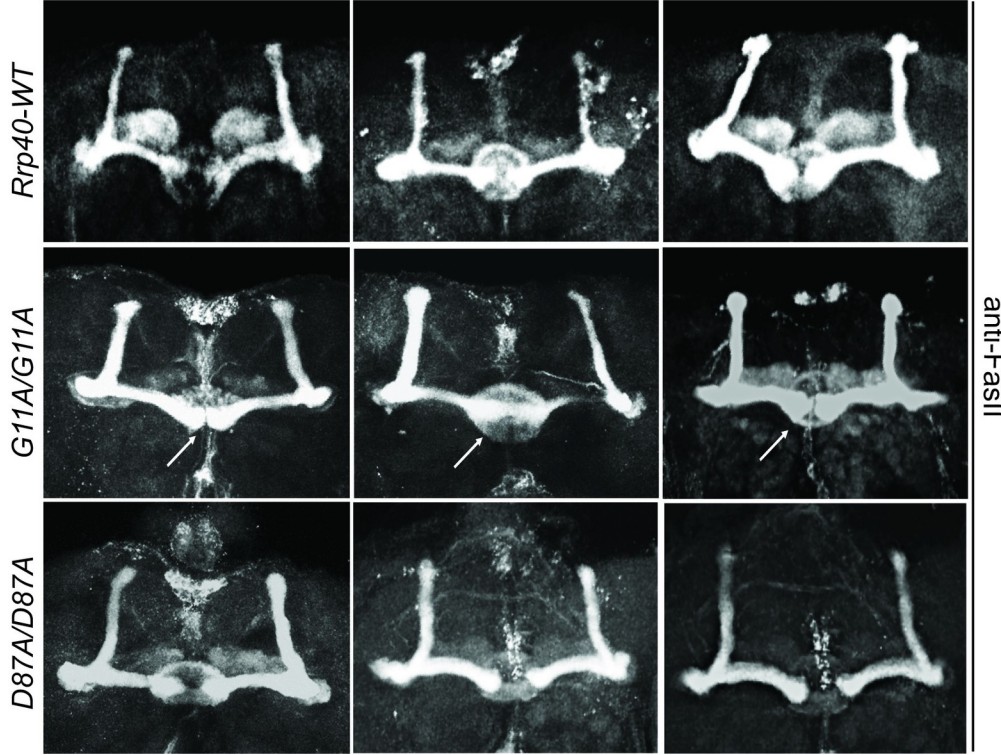

**Fig 5. $Rrp40^{G11A}$ mutant flies show defects in mushroom body morphology.** Three representative images of merged projections of FasII-stained mushroom bodies from *Rrp40-WT* control flies (n = 18) (top row), $Rrp40^{G11A}$ (G11A/G11A) mutant flies (n = 23) (middle row), and $Rrp40^{D87A}$ (D87A/D87A) mutant flies (n = 18) (bottom row) are shown. In control flies, β-lobes do not cross the mid-line and well-formed α and β-lobes are readily apparent. In $Rrp40^{G11A}$ mutant flies, 13/23 (57%) brains examined showed defects in β-lobe structure or β-lobe fusion (white arrows). Neither β-lobe defect was detected in the $Rrp40^{D87A}$ mutant flies.

**A**

| Model | Genotype | % eclosed (of expected) |
|---|---|---|
| WT | *Rrp40^wt^/Rrp40^wt^* | 93% n=100 |
| G11A | *Rrp40^G11A^/Rrp40^G11A^* UAS-EXOSC3-myc (rescue control) | 22% n=100 |
| | *Rrp40^G11A^/Rrp40^G11A^* elav-Gal4;UAS-EXOSC3-myc (pan-neuronal rescue) | 20% n=100 |
| | *Rrp40^G11A^/Rrp40^G11A^* Mhc-Gal4;UAS-EXOSC3-myc (muscle rescue) | 16% n=100 |
| D87A | *Rrp40^D87A^/Rrp40^D87A^* UAS-EXOSC3-myc (rescue control) | 37% n=100 |
| | *Rrp40^D87A^/Rrp40^D87A^* elav-Gal4;UAS-EXOSC3-myc (pan-neuronal rescue) | 40% n=124 |
| | *Rrp40^D87A^/Rrp40^D87A^* Mhc-Gal4;UAS-EXOSC3-myc (muscle rescue) | 43% n=75 |

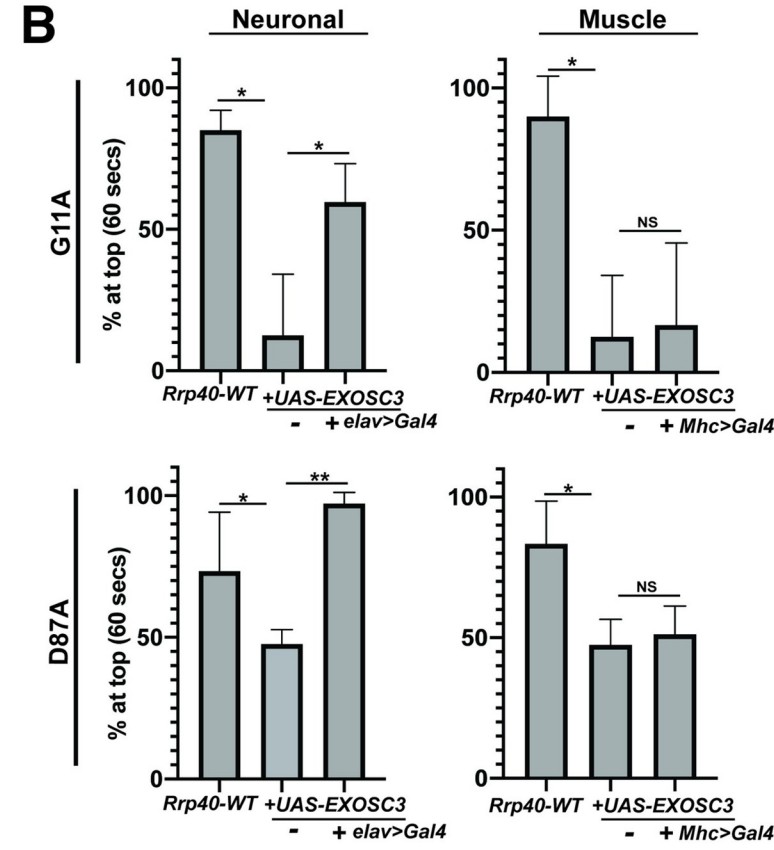

**Fig 6. Locomotor defects in *Rrp40* mutant flies are rescued by neuronal expression of human *EXOSC3*.** (A) Transgenic flies that express myc-tagged human *EXOSC3* under a UAS were generated as described in Materials and methods. The percentages of flies eclosed (of expected) for the indicated genotypes which include flies with transgenic *EXOSC3*, but no driver (rescue control), flies with the UAS and a pan-neuronal driver (*elav*) (pan-neuronal rescue), or the UAS and a muscle driver (*Mhc*) are shown for each *Rrp40* mutant. (B) Locomotor assays were employed to assess whether expression of human *EXOSC3* rescues behavioral phenotypes in *Rrp40* mutant flies (*G11A/G11A* or *D87A/D87A*) in either neurons (*elav*) or muscle (*Mhc*). Data are presented as the average percentage of flies that reach the top of a cylinder after 60 seconds across all trials. Groups of 10 age-matched flies (Day 6) were tested for at least three independent trials per genotype. Results are shown for control *Rrp40-WT* flies as compared to either *Rrp40^{G11A}* (*G11A/G11A*, top) or *Rrp40^{D87A}* (*D87A/D87A*, bottom) homozygous flies that either do not express human *EXOSC3* (-) or do express human *EXOSC3* (+Gal4) in either neurons (*elav*) (Neuronal, left) or muscle (*Mhc*) (Muscle, right). Values represent the mean ± SEM for n = 3 independent experiments. Asterisks (*) indicate results that show statistical significance at *p-value < 0.05; **p<0.01. Results that show no statistical significance when compared are indicated by NS.

indicating that each PCH1b-linked *Rrp40* allele alters the steady-state transcriptome relative to control *Rrp40^{wt}*. The transcriptomes of the two missense *Rrp40* mutants also differ from one another but not to same degree that either mutant differs from wildtype.

The high degree of reproducibility amongst the RNA-Seq replicates allowed us to identify transcripts that show statistically significant changes in each of the *Rrp40* mutant backgrounds relative to wildtype controls (**Fig 8B and 8C** and **S2** and **S3 Tables**). Consistent with the primary role of the RNA exosome in RNA decay [32], the majority of gene expression changes detected for *Rrp40^{G11A}* (**Fig 8B**) or *Rrp40^{D87A}* (**Fig 8C**) compared to *Rrp40^{wt}* are increases in steady-state RNA transcript levels. To illustrate this point, of the 385 transcripts that change 5-fold or more in the *Rrp40^{G11A}* mutant sample, 337 (88%) are increased (q<0.05), while of the 435 (84%) transcripts that change 5-fold or more in the *Rrp40^{D87A}* mutant sample, 366 (84%) (q<0.05) are increased. The total transcriptomic changes in the mutant *Rrp40* alleles show high correlation (R = 0.85) to one another based on comparing each *Rrp40* mutant to wildtype (**Fig 8D**). This analysis suggests a correlation in both the direction and magnitude of transcriptomic changes.

To provide some insight into the transcripts that are likely to be direct targets of the RNA exosome in this first analysis in neuronal tissues, we analyzed transcripts that show a 5-fold or more increase in steady-state levels. Of the total number of altered transcripts that change 5-fold or more, a majority are mRNAs (61% in *Rrp40^{G11A}* and 66% in *Rrp40^{D87A}*) and the remainder include several classes of ncRNAs (**Fig 8E**). The different classes of effected ncRNAs include lncRNA, snoRNA, asRNA, tRNA, scaRNA and sisRNA (**Fig 8F**). We assessed whether the *Rrp40^{G11A}* and *Rrp40^{D87A}* alleles show an increase in overlapping or distinct sets of RNAs (q<0.05, >5-fold change). Consistent with the correlation coefficient comparing overall expression changes (**Fig 8D**), we found many coordinately increased coding and non-coding transcripts between the two groups, but also some RNAs that were only increased >5-fold in one of the mutants (**S4 Fig**).

FlyEnrichr analysis (FlyEnrichr; amp.pharm.mssm.edu/FlyEnrichr/) of coding transcripts significantly increased (q<0.05, >1.2-fold change) in *Rrp40^{G11A}* and *Rrp40^{D87A}* heads identified enriched pathways linked to human disease, including frontotemporal dementia, amyotrophic lateral sclerosis, and neurodegenerative disease (**Fig 8G**).

## RNA-sequencing analysis reveals increased steady-state levels of key neuronal genes in *Rrp40* mutant flies

RNA-sequencing analysis revealed increased steady-state levels of multiple functionally important neuronal genes including *Arc1* [*Rrp40^{G11A}* (30-fold increase); *Rrp40^{D87A}* (16-fold increase)], *Or82a* [*Rrp40^{G11A}* (35-fold increase); *Rrp40^{D87A}* (2-fold increase)], *PNUTS*

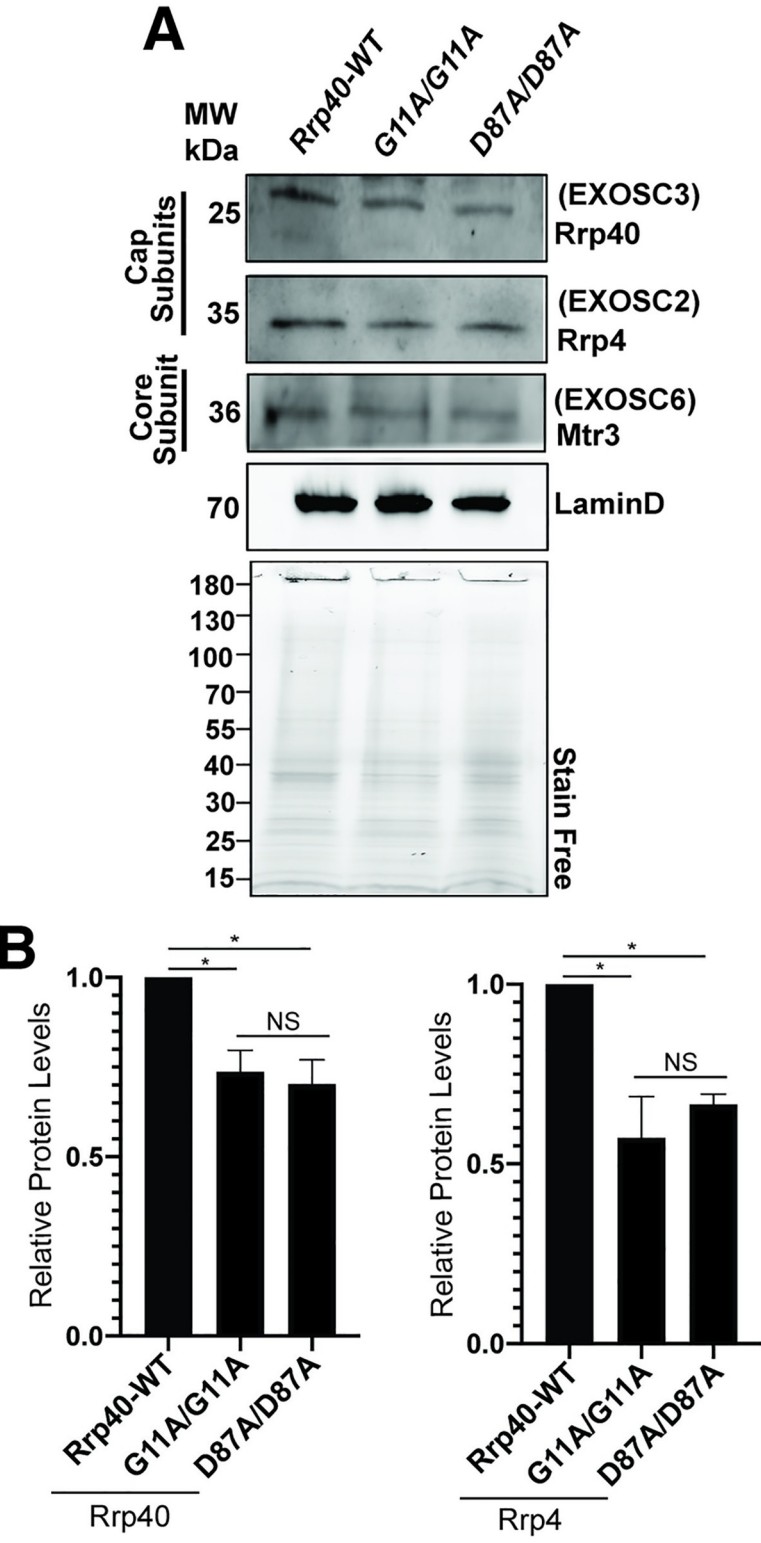

**Fig 7. Amino acid substitutions that model PCH1b in Rrp40 alter levels of RNA exosome subunits.** (A) Lysates prepared from heads of control *Rrp40^{wt}* or *Rrp40* mutant flies were resolved by SDS-PAGE and analyzed by immunoblotting with antibodies to detect RNA exosome Cap Subunits, Rrp40/EXOSC3 and Rrp4/EXOSC2, and Core Subunit Mtr3/EXOSC6. Both LaminD and Stain Free, as a measure of total protein, serve as loading controls. (B) Results from (A) were quantitated for RNA exosome cap subunits levels (Rrp40 and Rrp4- bands for Mtr3 had too

much background to yield reproducible results) and are presented as Relative Protein Levels with the value from the control Rrp40-WT flies set to 1.0. Asterisks (*) indicate results that show statistical significance at *p-value< 0.05. Results that show no statistical significance when compared are indicated by NS.

[*Rrp40*^*G11A*^ (3-fold increase); *Rrp40*^*D87A*^ (3-fold increase)], *WFS1* [*Rrp40*^*G11A*^ (7-fold increase); *Rrp40*^*D87A*^ (6-fold increase)], and *knon* [*Rrp40*^*G11A*^ (24-fold increase); *Rrp40*^*D87A*^ (17-fold increase)] in *Rrp40* mutants (**Fig 9**). We show Integrated Viewer (IGV) tracks generated from the RNA-seq dataset for *Arc1* (**Fig 9A**) as well as *Or82a*, *PNUTS*, *WFS1* and *knon (CG7813)* (**S5A–S5D Fig**). To validate altered gene expression from RNA-seq analysis, qRT-PCR was performed. This analysis confirmed that steady-state levels of *Arc1* mRNA in *Rrp40* mutants

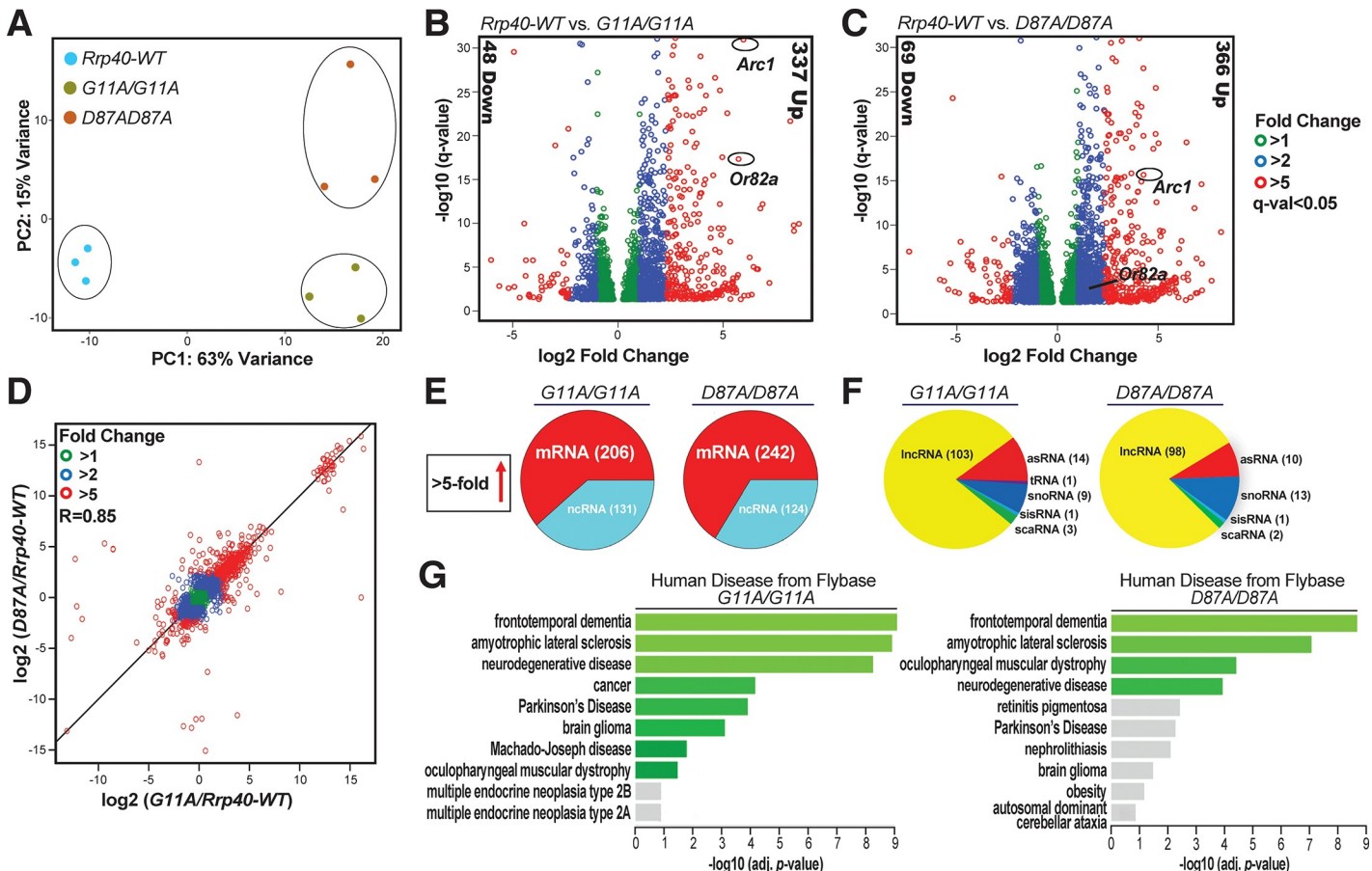

**Fig 8. RNA-seq analysis of *Rrp40* mutant flies identifies key neuronal targets that are altered.** (A) Principal component analysis (PCA) analysis of RNA-seq data from adult *Drosophila* heads from *Rrp40-WT* and *Rrp40* mutant flies (*Rrp40*^*G11A*^ and *Rrp40*^*D87A*^) shows that results from the three independent experiments cluster based on the genotype and that mutants are more distinct from control flies than from one another. (B,C) Volcano plots show transcripts differentially expressed in each *Rrp40* mutant [(B) G11A/G11A (2,351) and (C) D87A/D87A (3,502)] compared to *Rrp40*-WT (q<0.05, DEseq). The number of transcripts that show ≥ 5-fold change in steady-state levels (Down or Up) are indicated to the side in each plot. Representative regulators of neuronal function in flies (*Arc1* and *Or82a*) are highlighted. (D) A correlation curve comparing the changes in gene expression relative to wildtype for each *Rrp40* allele (*G11A/G11A* and *D87A/D87A)* was produced by plotting on a logarithmic scale. This analysis shows that the changes in transcript levels are highly correlated (R = 0.85) with respect to both magnitude and direction. (E) A pie chart illustrates the class of RNAs affected in each *Rrp40* mutant by showing total RNAs increased at least 5-fold in each *Rrp40* mutant corresponding to coding (mRNA) and non-coding RNA (ncRNA). (F) A pie chart illustrates the classes of ncRNAs increased at least 5-fold in each *Rrp40* mutant (lncRNAs, long noncoding RNAs; asRNAs, Antisense RNAs; tRNAs, transfer RNAs; snoRNAs, small nucleolar RNAs; sisRNAs, stable intronic sequence RNAs; scaRNAs, small Cajal body RNAs). (G) The enriched pathways from FlyEnrichr database for Human Disease from Flybase are shown for transcripts that are increased at least 1.2-fold in each *Rrp40* mutant (left, *G11A/G11A*; right, *D87A/D87A*) as compared to *Rrp40-WT*. The bars shown in green correspond to significant enrichment (Adj. p-val<0.05).

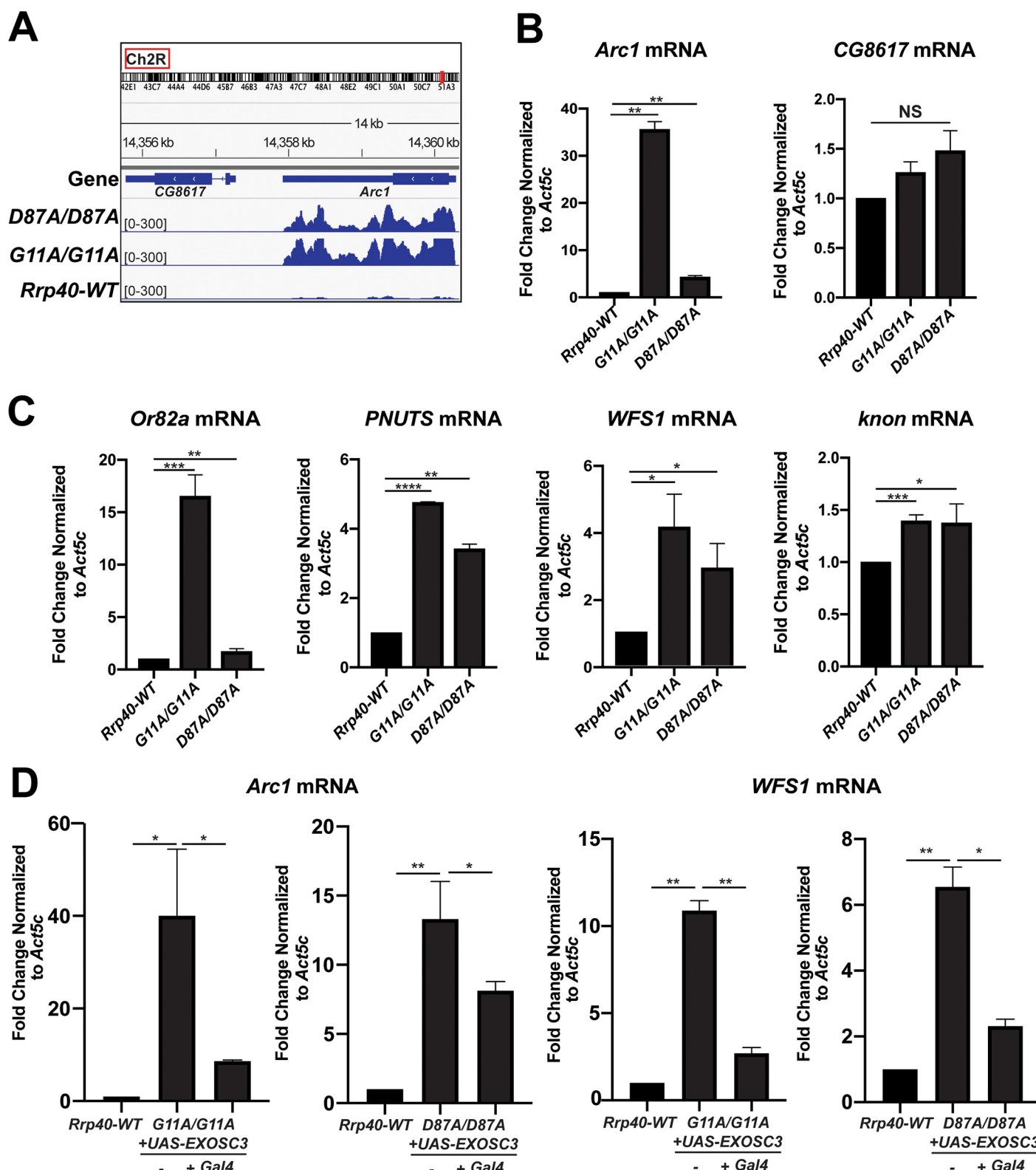

**Fig 9. Key neuronal transcripts show an increase in steady-state levels in *Rrp40* mutant flies.** (A) An Integrative Genomic View (IGV) screenshot (http://software.broadinstitute.org/software/igv/) of *Arc1* track peaks across the *Drosophila* genome (dm6 assembly) in *Rrp40* mutants [*Rrp40 D87A/D87A* (Upper) and *Rrp40 G11A/G11A* (Middle)] as well as *Rrp40-WT* (WT, Lower). Chromosome number and location (red rectangle; top) are displayed at the top. (B,C) Changes in the level of representative transcripts were validated using RT-qPCR in *Rrp40-WT* and *Rrp40* mutant flies. Data are presented as the Fold Change Normalized to *Act5c* transcript

(which was unchanged in the RNA-Seq datasets) where the value for the *Rrp40-WT* sample is set to 1.0. Results are presented as mean ± standard error of the mean (*p<0.05;**p<0.01;***p<0.001;****p<0.0001 vs. the WT group in a two-tailed *t*-test). (D) Expression of human *EXOSC3 (UAS-EXOSC3-myc)* in neurons rescues the increased steady-state levels of the *Arc1* (left) and *WFS1* (right) transcripts in both *Rrp40* mutants (*G11A/G11A* or *D87A/D87A*). The steady-state levels of the *Arc1* and *WFS1* transcripts were analyzed using RT-qPCR in control *Rrp40-WT* flies (set to 1.0) and mutant flies expressing human *EXOSC3* (+*Gal4*) or not (-). Data are presented as the means ± standard error of the mean (*p<0.05;**p<0.01 vs. the WT group in a two-tailed *t*-test).

(*Rrp40$^{G11A}$* and *Rrp40$^{D87A}$*) id significantly increased as compared to control (*Rrp40$^{wt}$*) (**Fig 9B**). As an additional measure of validation by RT-qPCR, we analyzed *CG8617* (**Fig 9B**), a neighboring gene of *Arc1* that showed no change in RNA-seq reads (**Fig 9A**). To extend the validation, we analyzed target transcripts increased in both *Rrp40* mutants compared to *Rrp40$^{wt}$*, including *Or82a*, *PNUTS*, *WFS1* and *knon* (**Fig 9C**). These results validate our RNA-seq analysis as well as highlight transcripts that may be potential regulators of neuronal function, providing candidates to help define the molecular mechanism for the phenotypes observed in both *Rrp40* mutants. Next, we tested whether expression of wildtype human *EXOSC3 (UAS-EXOSC3-myc)* only in neurons could rescue increased steady-state levels of validated transcripts, *Arc1* and *WFS1*. RT-qPCR confirmed that steady-state levels of these transcripts are significantly decreased compared to control UAS alone (-), when *EXOSC3* is expressed in neurons using a *Gal4* driver (+*Gal4*) (**Fig 9D**), providing evidence for molecular rescue by human EXOSC3 to complement rescue of behavioral phenotypes (**Fig 6B**).

## Discussion

In this study, we generated a *Drosophila* model to explore the requirement for RNA exosome function in specific tissues/cell types and define the functional consequences of amino acid substitutions in structural subunits of this complex that are linked to human disease. We provide evidence that the *Drosophila* Rrp40 subunit is essential for a critical step in early development. We also define an age-dependent requirement for Rrp40 in neurons critical for later stages of development and adult homeostasis. We extended this analysis to model genotypes that occur in PCH1b. These *Rrp40* missense mutations cause reduced viability and produce neuronal-specific behavioral phenotypes that align with PCH1b severity, revealing a genotype-phenotype correlation. The *Rrp40$^{G11A}$* mutant flies show defects in mushroom body morphology, highlighting the neurodevelopmental requirement for Rrp40. RNA-sequencing of the brain-enriched transcriptome of *Rrp40* mutant flies identifies specific changes in both coding RNAs, including the synaptic regulator *Arc1*, and non-coding RNAs. Consistent with impaired RNA exosome activity the majority of transcripts that change show an increase in steady-state levels. These data establish a critical role for the RNA exosome in neurons and provide an initial characterization of the functional consequences of amino acid changes that underlie PCH1b *in vivo* using a multicellular organism.

Previous studies of the RNA exosome have provided evidence that individual subunits of this complex are essential in the organisms analyzed [1, 2, 10–12, 14]. Thus, identification of mutations in genes encoding structural subunits of the RNA exosome that cause tissue-specific pathology was unexpected. In fact, only a few studies have explored the requirement for the RNA exosome in any specific tissues or cell types [12, 13, 33–35]. Prior work using methods to deplete individual subunits of the complex in *Drosophila* revealed that these subunits are essential [12, 36]. More recent work examining the role of the EXOSC3 subunit in human embryonic stem cells built on studies that have implicated the RNA exosome in restraining differentiation [33–35]. If there are specific requirements for the RNA exosome that are most critical for the development and/or homeostasis of certain cells or tissues, this could explain why mutations that impair the function of this complex preferentially affect certain tissues.

Indeed, our studies exploiting *Drosophila* uncover an age-dependent requirement for Rrp40 in neurons. However, mutations in multiple genes that encode structural subunits of the RNA exosome termed exosomopathies, including two cap subunit genes, *EXOSC3* [6, 21, 37–39] and *EXOSC2* [5], and two core subunit genes, *EXOSC8* [3] and *EXOSC9* [40, 41], have now been linked to genetic diseases with a variety of clinical presentations reported. The finding that multiple mutations that likely impair the function of the RNA exosome cause different clinical consequences is not consistent with a simple model where a defined tissue or cell type is highly dependent on the function of this complex. Rather the missense mutations that have been identified in multiple RNA exosome subunit genes likely impair tissue-specific functions of this complex. Indeed, although the number of patients for any given exosomopathy is small, there are now multiple missense mutations that have been identified [15], raising the questions of whether different amino acid changes that occur within the same RNA exosome gene impair the function of the complex in the same manner or to the same extent. A combination of approaches can be employed to understand how the amino acid changes that occur in patients affect the function of the RNA exosome.

This study uncovered at least two requirements for Rrp40. As with other RNA exosome subunits [6, 12, 13], depletion of Rrp40 globally or in multiple tissues causes complete lethality. This reveals an essential role for the RNA exosome in early development. When we bypass this early requirement, depletion of Rrp40 in neurons causes a decreased lifespan, which is not observed for other cell types. Modeling PCH1b mutations in *Rrp40* allowed us to assess age-dependent functional consequences of these amino acid changes. These PCH1b model flies show a number of behavioral and morphological changes linked to altered neuronal function [31, 42]. Consistent with predicted neuronal dysfunction, depletion or mutation of *Rrp40* is sufficient to cause morphological defects in MBs including the loss of α-lobes and defective β-lobe axon extension. These defects argue that Rrp40 function is required for proper axon guidance and neurite extension within the brain.

We employed the *Gal4-UAS* system to demonstrate that neuronal expression of human *EXOSC3* is sufficient to rescue climbing deficits and molecular phenotypes in *Rrp40* mutants. In contrast to expression in neurons, expression of *EXOSC3* in muscle does not rescue climbing defects (**Fig 6B**). Neuronal expression of *EXOSC3* could also restore multiple transcripts that increase in *Rrp40* mutant flies to normal levels. Interestingly, while locomotor function is rescued in *Rrp40* mutants by neuronal-specific expression of *EXOSC3*, viability is not. These findings suggest that there is an early developmental requirement for *Rrp40* that *EXOSC3* cannot replace; however, *EXOSC3* is sufficient to replace *Rrp40* in adult homeostasis.

There are multiple mechanisms by which a single amino acid substitution could impair RNA exosome function (**Fig 10**). Ultimately, any of these mechanistic consequences would alter RNA exosome activity toward target RNAs. Changes in all or a subset of these RNAs could cause pathology.

The wildtype RNA exosome associates with cofactors to facilitate both decay and processing of target RNAs (**Fig 10A**) [32]. In the mutant case, levels of the individual subunit or the overall assembled complex could decrease (a). In fact, a previous study using patient fibroblast and myoblasts to examine variants of EXOSC8, an RNase (PH)-like domain-containing core 'ring' subunit linked to another form of PCH [40, 41], not only showed reduced levels of the EXOSC8 variants, but also EXOSC3, indicating that defects in core 'ring' subunits of the RNA exosome can lead to a reduction in other subunits [3]. This finding is rather surprising given that the patients analyzed did not exhibit pathology related to the cell types analyzed. While our data show a modest decrease in levels of Rrp40 and an associated cap subunit Rrp4 (**Fig 7**), two lines of evidence suggest this decrease is not sufficient to explain the phenotypes reported in this study. First, both *Rrp40* alleles examined show a similar decrease in levels of Rrp40 as

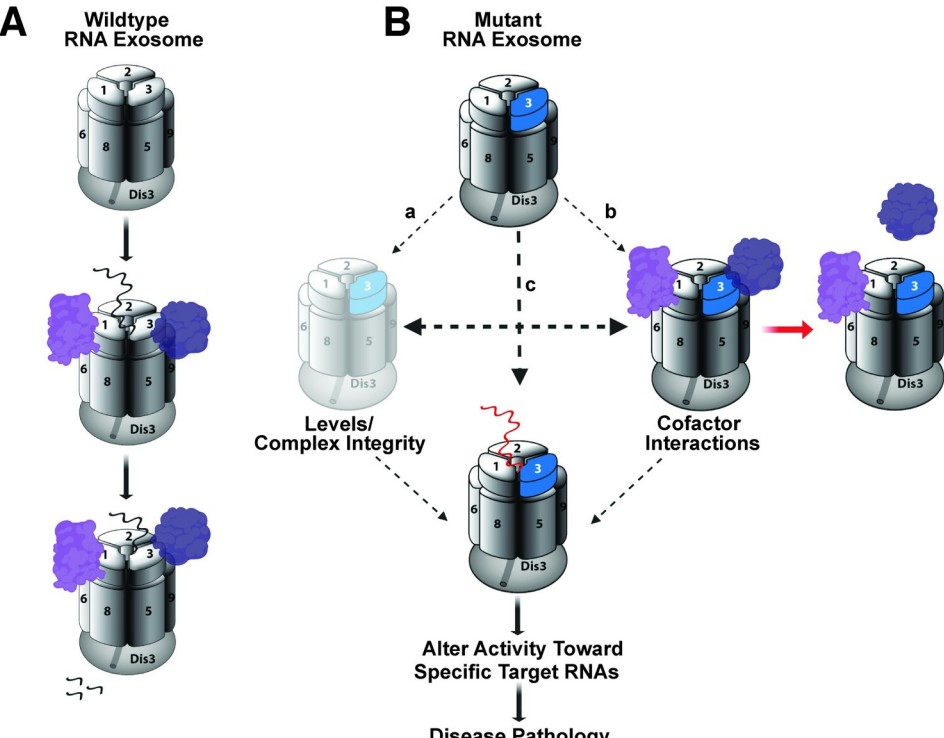

**Fig 10. Mechanistic model for how amino acid substitutions could alter RNA exosome function.** (A) The Wildtype RNA exosome (top) associates with a variety of cofactors (purple) that can aid in target RNA recognition (middle) and decay/processing (bottom). (B) Amino acid substitutions that occur in the EXOSC3/Rrp40 cap subunit (shown in blue) could impact the function of the complex through multiple mechanisms, including: (a) decrease the steady-state level of the specific subunit or alter interaction of the subunit with the complex, leading to a decrease in overall Levels or altered Complex Integrity; (b) affect Cofactor Interactions, which could impact interactions with target RNAs; and/or (c) could directly alter interactions with target RNAs (red RNA indicates impaired interaction). Ultimately, any of these changes could Alter Activity Toward Specific Target RNAs. Changes in the steady state level or processing of RNAs could then cause downstream Disease Pathology.

well as Rrp4, but *Rrp40^{G11A}* mutant flies consistently show more severe phenotypes than *Rrp40^{D87A}* flies (**Figs 3B, 3C, 4 and 5**). Furthermore, analysis of flies with a deficiency that removes the *Rrp40* gene, creating flies with a single copy of *Rrp40*, do not display any of the phenotypes detected in the engineered *Rrp40* mutant flies. These deficiency flies should have a 50% decrease in levels of Rrp40, which is a larger decrease than we detected in either *Rrp40* mutant fly (**Fig 7B**).

In addition to overall loss of the complex, amino acid changes in the RNA exosome could alter interactions with cofactors (b). Indeed, structural studies of the budding yeast RNA exosome show that the nuclear RNA exosome cofactor Mpp6 interacts with the Rrp40 cap subunit [43, 44], and that an amino acid change in Rrp40 that models a change found in PCH1b decreases the interaction of Mpp6 with the RNA exosome [44]. The idea that interactions with specific cofactors could be perturbed by the amino acid changes that occur in disease is attractive because altered interactions with cofactors in different tissues or cell types could explain the phenotypic variation observed for mutations within the same *EXOSC* gene or the different *EXOSC* subunit genes that are linked to distinct diseases. RNA exosome cofactors have not been studied in *Drosophila*. The *Rrp40* mutant flies described here could be a valuable tool for future genetic studies to define such cofactors.

Given that EXOSC3/Rrp40 is a cap subunit, amino acid substitutions could also directly impair interactions with target RNAs (c). In budding yeast, the three cap subunits of the RNA exosome make direct contact with target RNAs [9, 43], making this mechanistic consequence a formal possibility, Future studies will be required to understand how the different amino acid changes that occur in PCH1b impair the function of the RNA exosome.

As a first approach to consider how amino acid substitutions in EXOSC3 could impact the overall RNA exosome complex, we can take advantage of the availability of structures of this large macromolecular complex [8, 9, 43, 45, 46]. While there is no structure available of the *Drosophila* RNA exosome, the considerable evolutionary conservation illustrated in **Fig 3B** provides some context to consider how distinct amino acid changes in EXOSC3/Rrp40 could impair RNA exosome function. As shown in **Fig 3A**, the amino acid changes modeled in this study lie within the N-terminal (G31A) and S1 (D132A) domains of EXOSC3/Rrp40. The G31 residue in EXOSC3 packs against the surface of EXOSC5 and, therefore, the EXOSC3-G31A substitution could impair the interaction of EXOSC3 with EXOSC5 [8]. The D132 residue in EXOSC3 is located in a loop between strands in the S1 domain and the D132A substitution could thus impair the folding of the loop and, subsequently, disturb interactions between EXOSC5 and EXOSC9 [8]. These observations suggest that PCH1b amino acid changes in EXOSC3 could impair interactions with other RNA exosome subunits, leading to compromised RNA exosome complex integrity; however, wholesale loss of the RNA exosome seems unlikely to explain the phenotypic variability reported in PCH1b [47].

Here, we focused on defining the spectrum of RNA targets that are affected when Rrp40 function is impaired. To address this aspect of the mechanistic model (**Fig 10B**), we performed RNA-Seq analysis, providing the first dataset of candidate RNA exosome targets in brain tissue. Our brain-enriched RNA-sequencing data on *Rrp40* mutant flies shows a global increase in transcripts and specific differences in both coding and non-coding transcripts in each *Rrp40* mutant compared to control (**Fig 8B and 8C**). The finding that more RNAs show an increase in steady-state levels as compared to a decrease is consistent with impairing the function of a complex required for RNA turnover. Our results show significant correlation between the transcriptomic changes in *Rrp40*[G11A] flies and *Rrp40*[D87A] flies (**Fig 8D**), but also some clear differences both in the identity of the transcripts effected (**S4 Fig**) and the magnitude of the effects (**Fig 9**). The specific transcripts altered, the magnitude of the changes in the same transcripts or sets of transcripts, or a combination of these effects could underlie the phenotypic differences in the *Rrp40* mutant flies analyzed. Future studies, including additional models, will be required to define how changes in the transcriptome contribute to these phenotypic differences.

Levels of a critical synaptic regulator, *Arc1*, are markedly increased in both *Rrp40* mutants, suggesting that the RNA exosome regulates key neuronal transcripts. Notably, *Arc1* levels increase more in the *Rrp40*[G11A] flies compared to *Rrp40*[D87A], consistent with a model where the magnitude of changes in key regulators could contribute to phenotypic variation. Critically, these data provide insight into how amino acid changes in Rrp40 could disrupt processing and/or decay of transcripts required for proper development/maintenance of the nervous system.

In this study, we modeled three patient genotypes associated with PCH1b (**Fig 3B**). The analysis reveals a range of phenotypes, suggesting a genotype-phenotype correlation. Furthermore, the spectrum of phenotypes observed in flies modeling alleles of PCH1b aligns with some aspects of genotype-phenotype correlations reported for individuals with PCH1b, albeit the clinical phenotype are highly variable [39, 47], suggesting that these amino acid changes differentially impact RNA exosome activity. For example, homozygous PCH1b *EXOSC3* p. Gly31Ala mutations [6, 20, 39], modeled as *Drosophila Rrp40*[G11A], cause severe phenotypes

(Fig 4). In contrast, homozygous PCH1b *EXOSC3* p.Asp132Ala mutations, modeled as *Drosophila Rrp40*$^{D87A}$, cause more moderate phenotypes as compared to the *Rrp40*$^{G11A}$ mutant flies. Moreover, PCH1b *EXOSC3* p.Asp132Ala *in trans* to a presumptive *EXOSC3* null mutation, modeled as *Drosophila Rrp40*$^{D87A/Df(2L)}$ causes the most severe patient phenotypes [6], arguing that *D132A* is likely hypomorphic. Interestingly, *EXOSC3* p.Gly31Ala *in trans* to a presumptive *EXOSC3* null has not been reported, and the lethality of the corresponding genotype *Rrp40*$^{G11A/Df(2L)}$ in flies (Fig 3B) suggests that human G31A and *Drosophila* G11A are strong loss of function alleles. While the fly models can provide insight into genotype-phenotype correlation, it is important to note that pathology and/or disease severity could arise from a combination of effects, and these combined effects could be specific to the genetic background of the patients. Critically, the number of reported individuals with *EXOSC3*-related PCH1b is small [6, 20, 37–39, 48], thus limiting the ability to make genotype-phenotype correlations in patients and highlighting the value of the fly model.

In summary, this study identifies an essential role for Rrp40 in early development in *Drosophila* together with an age-dependent requirement for Rrp40 in neurons. Using CRISPR/Cas9 gene editing technology and *Drosophila* genetic approaches, we modeled PCH1b-linked EXOSC3 amino acid changes in Rrp40, which reveals neuronal-specific phenotypes and defines RNA targets critical for neuronal function. Strikingly, both behavioral phenotypes of *Rrp40* mutant flies and levels of RNA targets are rescued by pan-neuronal transgenic expression of human *EXOSC3*. Taken together, these results demonstrate that this model can be utilized to explore the broader consequences of amino acid changes in the RNA exosome.

## Materials and methods

### *Drosophila* stocks

Crosses were maintained in standard conditions at 25˚C unless otherwise noted. Stocks used in this study: *w*$^{1118}$, *UAS-Rrp40*$^{RNAi}$ (TRiP *HMJ23923*, BDSC #63834), *Df(2L)Exel6005* (*BDSC #7492)*, *Tub>Gal80*$^{ts}$ (*BDSC #7017*), and the following Gal4 stocks: *Actin5c>Gal4 (BDSC#25374)*, *elav>Gal4 (BDSC#458)*, *repo>Gal4 (BDSC#7415)* and *Mhc>Gal4 (BDSC#55133)* were obtained from the Bloomington *Drosophila* Stock Center. *Rrp40*$^{G11A}$ and *Rrp40*$^{D87A}$ were generated by CRISPR/Cas9 editing (detail below) at Bestgene, Inc. (CA).

### Brain dissections and immunohistochemistry

Brain dissections and staining were performed as described previously [49]. Briefly, brains of anesthetized animals were dissected in PTN buffer (0.1M NaPO$_4$, 0.1% Triton X-100), fixed in 4% paraformaldehyde (Electron Microscopy Sciences), and then stained overnight with primary antibody (ID4) diluted in PTN. Following several washes, brains were incubated with the appropriate fluorescently conjugated secondary antibody (1:250) in PTN for 3 hours at room temperature, washed in PTN, and then mounted in Vectashield (Vector Labs). The 1D4 anti-FasII hybridoma (1:20) developed by C. Goodman [50] was obtained from the Developmental Studies Hybridoma Bank (DSHB).

### Generation of CRISPR/Cas9 flies

**Molecular reagents.**   *pU6-gRNAs*: Target-specific sequences were identified sequences using DRSC flyCRISPR optimal target finder (https://www.flyrnai.org/crispr/) for Rrp40 and gRNAs were synthesized as 5'-phosphorylated oligonucleotides, annealed, and ligated into the *Bbs*I sites of pU6-*Bbs*I chiRNA [51].

*Homology-directed repair (HDR) templates*: HDR donor vectors (Emory Integrated Genomic Core) were constructed by cloning a 3kb fragment of the *Rrp40* locus, including a loxP-flanked *3xP3-DsRed* cassette inserted downstream of the *Rrp40 3'UTR*, into *KpnI/SmaI* sites of the *pBlueScript-II* vector. Base changes corresponding to G11A and D87A were engineered into this backbone. The *3xP3-DsRed* cassette allows positive selection based on red-fluorescence in the adult eye [52].

*CRISPR-transformants*: Injection of DNA mixtures (500 ng/µl HDR and 250 ng/µl U6-gRNA plasmid) into *nos-Cas9* embryos and subsequent screening for dsRed+ transformants was performed by Bestgene, Inc. CRISPR. Edits were confirmed by direct sequencing (S3 Fig).

## Lifespan analysis

Lifespan was assessed at 25˚C as described previously [53]. In brief, newly eclosed animals were collected, separated by sex, placed in vials (up to 20 per vial), and transferred to fresh vials weekly. Survivorship was scored daily. At least 8 vials of each genotype were tested and averaged to calculate mean and standard deviation. Log-rank tests of survivorship curves were performed using PRISM (GraphPad, San Diego).

## Behavioral Assays

Negative-geotaxis assay was performed as previously described [54] with some modification. Newly eclosed flies (*Rrp40$^{wt}$*, *Rrp40$^{G11A}$*, *Rrp40$^{D87A}$*) (day 0) were collected, divided into groups of 10, and kept in separate vials. Cohorts of age-matched flies were then transferred to a 25-ml graduated for analysis of climbing response. At least 10 groups (i.e. ~100 flies) were analyzed per genotype.

## Immunoblotting

Protein lysates of whole flies or heads were resolved on 4–20% Criterion TGX polyacrylamide gels (Bio Rad), transferred to nitrocellulose membranes, incubated for ≥1hr in blocking buffer (5% non-fat dry milk in 0.1% TBS-Tween), followed by an overnight incubation at 4˚C in primary antibody diluted in blocking buffer. Primary antibodies were detected using species-specific horse radish peroxidase (HRP) conjugated secondary antibodies (Jackson ImmunoResearch) and enhanced chemiluminescence (ECL, Sigma). Primary antibodies include: guinea pig anti-Rrp4 (1:1000), rabbit anti-Rrp40 (1:1000), guinea pig anti-Csl4 (1:000), and rat anti-Mtr3 (1:1000) gift of the Lis lab [36, 55].

## RNA sequencing analysis

RNA-seq was performed on three replicates of 60 newly-eclosed adult heads per genotype (*Rrp40$^{G11A}$*, *Rrp40$^{D87A}$* and *Rrp40$^{wt}$*). Heads were collected and lysed in TRIzol (Thermo-Fisher). rRNA was depleted and cDNA libraries were prepared using Truseq RNA sample prep kit (Illumina). The cDNA paired-end reads were sequenced at the Georgia Genomics and Bioinformatics Core at UGA using a NextSeq (150 cycles) PE75 High Output Flowcell. The reads were mapped to the BDGP *D. melanogaster* (r6.25) genome. Several sequence datasets were downloaded from GenBank and FlyBase for initial assessment of the data including: 1) All Drosophila ribosomal sequences found in Genbank: *Drosophila*_rRNA.fa; 2) All FlyBase *D. melanogaster* transcripts from assembly r6.25: dmel_transcripts_r6.25.fa. Adapters were trimmed from raw reads using Trimmomatic (ver. 0.36) [56], and trimmed reads were aligned

to the ribosomal sequences using Bowtie2 [57], and only reads which did not map to the ribosome were kept for downstream analysis.

Non-ribosomal reads were then mapped to D. melanogaster r6.25 using Tophat2 [58]. The resulting aligned reads were compared using DESeq2 [59] to identify genes that change significantly (p-value<0.05, >1.5-fold change) in expression. Only genes that were significantly changed compared to the control CRISPR line, $Rrp40^{WT}$, were used for further analysis.

## RNA isolation and qPCR

Total RNA was isolated from heads with TRIzol (Invitrogen) and treated with Turbo DNase (Invitrogen) to degrade contaminating DNA. For qRT-PCR analyses, cDNA was generated using MMLV Reverse transcriptase (Invitrogen) from 1 μg of total RNA, and then used as template for quantitative real-time PCR (qPCR) from duplicate samples of 10 ng cDNA with QuantiTect SYBR Green Master Mix using an Applied Biosystems StepOne Plus real time machine (ABI). Results were analyzed using the ΔΔCT method and normalized to *Act5c* as fold change relative to control. Oligonucleotide sequences are provided in (**S1 Table**).

## Statistical analysis

All statistical analyses were performed in GraphPad (San Diego, CA). Comparisons between experimental groups were made using one-way analysis of variance (ANOVA), unless noted otherwise. All data are presented as means and standard error of the mean (SEM) (error bars) for at least three independent experiments. Asterisks (*) indicate statistical significance at p-value < 0.05.

## Supporting information

**S1 Fig. Validation of Rrp40 RNAi.** An immunoblot of protein extracts from heads of adult flies expressing *elav-Gal4* (neuronal) either without (-) or with (+) an RNAi targeting *Rrp40* (*elav-Gal4>UAS-Rrp40^{IR}*) probed with α-Rrp40 antibody [55] is shown. The α-Rrp40 antibody detects Rrp40 at the predicted size (25 kDa) and this band is efficiently depleted in flies the express the UAS-RNAi. A non-specific band (*) detected by the antibody is not affected. Ponceau staining provides total protein as a loading control. Flies were reared at a lower temperature (18˚C) to recover viable animals for analysis.
(TIFF)

**S2 Fig. CRISPR/Cas9-induced Homology-directed recombination (HDR) with a double-stranded DNA donor vector.** (A) Schematic of the HDR strategy used to replace the *Rrp40* locus with the edited *Rrp40-3xP3-DsRed* vector, which drives expression of DsRed in the eye. DsRed is flanked by loxP recombination sites to remove DsRed if desired. The guide RNA (gRNA) target site is indicated (red arrow). Homology arms of 1 kb immediately flanking *Rrp40* were cloned into the *Rrp40-3xP3-DsRed* vector with the desired mutations engineered into the *Rrp40* gene to produce the edited ORF.
(TIFF)

**S3 Fig. *EXOSC3*-PCH1b mutations modeled in *Drosophila Rrp40*.** (A,B) Sequences confirming the creation of CRISPR/Cas9 genome-edited flies (A) $Rrp40^{D87A}$ and (B) $Rrp40^{G11A}$ isolated are shown. The engineered mutations in the *Rrp40* locus are highlighted in red and are noted along with sequencing traces of (C) $Rrp40^{D87A}$ and (D) $Rrp40^{G11A}$ with the engineered mutations indicated by a red box, confirming the successful generation of the edited alleles.
(TIFF)

**S4 Fig. Venn diagrams to illustrate the overlap in transcripts that increase by 5-fold or more in *Rrp40^{G11A}* and *Rrp40^{D87A}* mutant flies compared to wildtype.** Venn diagrams illustrate the overlap of protein-coding (mRNA) and non-coding (ncRNA) transcripts using more than a 5-fold increase as cut off, comparing the *G11A/G11A* and *D87A/D87A Rrp40* mutant flies to one another.
(TIFF)

**S5 Fig. Integrative Genomic View (IGV) screenshots of functionally important neuronal transcripts to illustrate the RNA-Seq data obtained.** Integrative Genomics Viewer (IGV) screenshot (http://software.broadinstitute.org/software/igv/) for (A) *WFS1*, (B) *Or82a*, (C) *PNUTS* and (D) *knon (CG7813)* track peaks across the *Drosophila* genome (dm6 assembly) in *Rrp40* mutants [*Rrp40^{D87A}* (D87A/D87A, Upper) *Rrp40^{G11A}* (G11A/G11A, Middle)] as well as *Rrp40^{wt}* (Rrp40-WT, lower).
(TIFF)

**S1 Table. Oligonucleotide primer sequences employed for RT-qPCR.**
(TIF)

**S2 Table. RNA-Seq data analysis of *Rrp40^{G11A}* alleles.**
(XLSX)

**S3 Table. RNA-Seq data analysis of *Rrp40^{D87A}* alleles.**
(XLSX)

**S1 Video. Negative geotaxis assay for *Rrp40^{wt}* at Day 4.**
(MOV)

**S2 Video. Negative geotaxis assay for *Rrp40^{G11A}* at Day 4.**
(MOV)

**S3 Video. Negative geotaxis assay for *Rrp40^{D87A}* at Day 4.**
(MOV)

## Acknowledgments

We thank the members of the Corbett and Moberg laboratories for their intellectual contributions. We also thank Dr. Ambro van Hoof for careful reading of the manuscript and critical suggestions. The RNA-sequencing service was provided by the Georgia Genomics and Bioinformatics Core. We thank the Georgia Genomics and Bioinformatics Core for technical support.

## Author Contributions

**Conceptualization:** Derrick J. Morton, Sara W. Leung, Milo B. Fasken, Kenneth H. Moberg, Anita H. Corbett.

**Data curation:** Derrick J. Morton.

**Formal analysis:** Derrick J. Morton, Isaac Kremsky.

**Funding acquisition:** Derrick J. Morton, Kenneth H. Moberg, Anita H. Corbett.

**Investigation:** Derrick J. Morton, Binta Jalloh, Lily Kim, Isaac Kremsky, Rishi J. Nair, Khuong B. Nguyen, J. Christopher Rounds, Maria C. Sterrett, Brianna Brown, Thalia Le, Maya C. Karkare, Kathryn D. McGaughey, Shaoyi Sheng.

**Methodology:** Derrick J. Morton, Kenneth H. Moberg, Anita H. Corbett.

**Project administration:** Derrick J. Morton, Anita H. Corbett.

**Resources:** Derrick J. Morton.

**Supervision:** Derrick J. Morton, Anita H. Corbett.

**Validation:** Derrick J. Morton.

**Visualization:** Derrick J. Morton.

**Writing – original draft:** Derrick J. Morton, Anita H. Corbett.

**Writing – review & editing:** Derrick J. Morton, Kenneth H. Moberg, Anita H. Corbett.

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
