## [Decision Letter · Decision Letter 0]

18 Oct 2019

Dear Dr Corbett,

Thank you very much for submitting your Research Article entitled 'A Drosophila Model of Pontocerebellar Hypoplasia Reveals a Critical Role for the RNA Exosome in Neurons' to PLOS Genetics. Your manuscript was fully evaluated at the editorial level and by independent peer reviewers. The reviewers appreciated the attention to an important problem, but raised some substantial concerns about the current manuscript. Based on the reviews, we will not be able to accept this version of the manuscript, but we would be willing to review again a much-revised version. We cannot, of course, promise publication at that time.

[GPC Note: One of the reviewers commented to the editors that raw RNAseq datasets were not available.  In compliance with PLOS Genetics open data policy we ask that you make sure to remedy this issue in your revision.]

If you decide to revise the manuscript for further consideration at PLOS Genetics, please aim to resubmit within the next 60 days, unless it will take extra time to address the concerns of the reviewers, in which case we would appreciate an expected resubmission date by email to plosgenetics@plos.org.

[LINK]

We are sorry that we cannot be more positive about your manuscript at this stage. Please do not hesitate to contact us if you have any concerns or questions.

Yours sincerely,

Hugo J. Bellen, Ph.D., D.V.M.

Guest Editor

PLOS Genetics

Gregory P. Copenhaver

Editor-in-Chief

PLOS Genetics

Reviewer's Responses to Questions

**Comments to the Authors:**

Reviewer #1: Morton et al. have investigated the neuronal requirements of EXOSC3 and the RNA exosome in a “Drosophila model of pontocerebellar hypoplasia (PCH1b)”. Recent evidence strongly implicates RNA metabolism in the pathogenesis of neurologic disorders and for neuronal maintenance. Given the expected broad requirement for exosome functions across all cell types, how genetic variants in this complex specifically affect neurons is an interesting unanswered question. Using a CRISPR strategy, Morton et al. introduce 2 PCH1b mutations into fly Rrp40, the ortholog of EXOSC3, and characterized resulting phenotypes affecting survival and locomotor impairment. Perhaps the most interesting and important finding is evidence to support an “allelic series” of phenotypic severity which shows some consistency with genotype-phenotype studies of human PCH1b patients. RNA-sequencing in the fly mutants also identify many transcripts showing increased abundance, consistent with altered stability/turnover, including in the synaptic regulator, Arc1.

Overall, this manuscript makes some important initial contributions to understanding the role of EXOSC3 in the adult brain and its potential impact on the transcriptome; however, it also leaves many questions unanswered, with 2 particular issues looming large. First, while the potential cross-species genotype-phenotype correlations are intriguing, the potential mechanism(s) of the PCH1b-causing variants remains unresolved. While the RNA-seq analyses confirm a global transcriptome perturbation, the allele with more severe phenotypes (G11A) has a less substantial impact on the transcriptome overall [number of transcripts affected and magnitude of change, Fig 6B vs. C]—this should be discussed. Notably, Arc1 does appear to be more severely affected in G11A, but this is correlative in the absence of additional evidence that Arc1 modifies Rrp40-dependent phenotypes. Since Rrp40 protein expression and that of several other exosome complex proteins appears unaffected (but see below), other in vivo or in vitro studies might be considered to provide clues as to the molecular mechanisms for these variants and the relative phenotypic strengths. Second, and perhaps even more importantly, since the manuscript purports to develop “A Drosophila model of pontocerebellar hypoplasia”, it is essential to determine if the model recapitulates the developmental and/or pathologic degenerative features that define this disorder. The phenotypes characterized here (survival, locomotor impairment, and wing posture) are rather non-specific. While the rescue studies support a neuronal origin, some further experiments might easily address whether the fly brain and motor neurons are normally developed and whether there is evidence of neurodegeneration with aging (e.g. adult brain histology in newly-eclosed and aged adults).

The following are additional suggestions to improve the manuscript.

Major critiques/questions:

-As noted above, studies should be performed to broadly survey structure and maintenance of adult brain structures to determine whether neurodevelopmental and/or degenerative aspects of PCH1b are recapitulated (e.g. studies in young and aged adults). It may also be informative to examine other accessible nervous system structures, including the larval neuromuscular junction, since spinal motor neuron degeneration is a defining feature of PCH1b (based on OMIM); quick surveys of the adult retina and larval brain may also be informative.

-The RNA-interference experiments appear to based on a single transgenic line, raising the possibility of off-target effects. Did the authors validate by western blot or RT-PCR that RNAi reduces Rrp40 expression?—this would also enhance interpretation of Fig 5.

-The table in fig. 4A appears to largely present negative data—the absence of rescue for semi-lethality, but this was not addressed further in the text. Some context and interpretation should be provided. Do ubiquitous GAL4 drivers (e.g. act5c) rescue viability? In a related question, did the authors consider attempting rescue of the locomotor phenotype using a PNS, motor neuron, or muscle specific driver line? Motor neuron might be especially meaningful given the involvement in PCH1b.

-Also alluded to above, western blot evidence is provided (Fig. 5) suggesting that the Rrp40 alleles created do not significantly affect protein expression; however, the data does not allow assessment of a more modest, but significant impact. This is an important experiment, and it would be advisable to perform several replicates and provide a quantification with statistical analysis. Also, have prior studies from human tissue or cell culture shown consistent results?—if yes this should be mentioned. This result might be considered to present earlier (with Fig 2).

-If the Rrp40 variants do not affect protein expression or stability, it would be informative to address other possible mechanisms. Are immunoprecipitation experiments feasible to address whether the complex is intact? Other possibilities are to examine localization (in vivo) or RNA-binding or other assay for activity (in vitro). At a minimum, the possible mechanisms should be thoroughly discussed, including future experimental directions that would be helpful to resolve.

-How did the authors choose to focus on the specific transcripts presented in Fig. 7 and in the results text (first paragraph of “steady state levels of key neuronal genes” section)? Without a systematic description, it appears that this subgroup was “cherry-picked”. Importantly, are there specific sequence features that suggest RNAs affected by Rrp40 alleles are targeted by the exosome? It appears that exosome may target RNAs containing AU-rich elements in 3’UTRs—can this be experimentally validated from the RNAseq data. Given the involvement of the exosome in nonsense mediated decay, one might expect the accumulation of transcripts containing errors, or secondary changes in other proteins that respond to transcriptional errors.

-Are there available data from human studies (either cells or postmortem tissue) that can inform on whether the RNA transcripts that change in Rrp40 mutants represent either conserved transcriptomic features of PCH1b or exosome failure. Are the genes disrupted in fly heads conserved and expressed in human neuronal tissues that are impacted in PCH1b? Is this first transcriptome-wide analysis of consequences following genetic disruption of the RNA exosome? If yes, this should be made clear, as it will increase interest. If not, than it is important to compare the transcripts affected with other available datasets.

Other issues for consideration:

-It would be helpful to include a bit more detail in the introduction describing the PCH1b disease phenotype, including developmental and degenerative aspects, and the specific observations that support a genotype-phenotype correlation—this is important to interpret the value of fly model.

-Please add to background or results the overall similarity and identity by amino acid sequence of Rrp40 in flies with human EXOC3.

-Was any analysis of heterozygous animals performed (G11A/+ or D87A/+) to exclude potential dominant negative effects?

-The significance of the “wing posture” phenotype is obscure. Can the authors provide citations to support the statement that this is “consistent with neurological deficits”?

-“Seizure-like behavior” is not immediately obvious in the provided video S1; It would be preferable to describe the behavior rather than use the descriptor “seizure-like”, which may not mean the same thing to all readers.

-Maybe mention the age of the adult flies used for RNA-seq studies in the results?

-The y-axis scale is highly variable in Fig 7 B-D. it would be more clear to show a common scale so the relative magnitude of RNA expression changes can be appreciated.

-It will be helpful to provide full genotypes in figure legends or methods. The genotypes for controls in some studies are not clear (Fig. 1E, 4B, 7D).

-There does not appear to be a section in methods or statements in figure legends adequately describing the statistical methods (e.g. Fig. 1E, 2B/C, 3B, and 4A). Were appropriate post-hoc tests used to adjust for multiple comparisons (e.g. Fig.7)?

-Please include scale bars for the IGV plots (Fig. 7A, S5).

-Please provide column headers to aide interpretation of Table S4.

Reviewer #2: Overview

In this manuscript, Morton et al characterize the in vivo function the Rpr40 gene in Drosophila, which encodes a conserved component of cap region of the RNA exosome complex. To study the roles of Rpr40, they first conduct in vivo RNAi experiments across different tissues, using the Gal4-UAS system, revealing a particularly strong deleterious effect of Rpr40 loss-of-function (LOF) in neuronal and glial cells on post-embryonic viability. Since the human orthologue of Rpr40, EXOSC3, has been implicated in the neurological disorder Pontocerebellar Hypoplasia Type 1b (PCH1b), using CRISPR/Cas9 technology they also created Prp40 variant fly models bearing PCH1b-associated mutations. These mutant strains have reduced viability and lifespan, with survivors displaying wing abnormalities and behavioural defects consistent with neurological impairments. Finally, their comparative transcriptomic analyses of mutant and wildtype flies identified several hundred upregulated mRNAs associated with gene regulatory functions (e.g. DNA binding, helicase activity, RNA binding), as well as several neuronal transcripts (e.g. Arc1, Or82A). Altogether, this study offers a characterization of new Drosophila models of PCH1b, while also revealing the tissue-type specific requirements for RNA exosome components.

General Assessment

Overall, this paper presents a well-executed set of experiments that employ gold-standard approaches (e.g. CRISPR genome editing) to address the in vivo functions of Prp40 in Drosophila. The finding that Prp40 LOF has a more severe effect in neurons and glial cells is certainly interesting. The study could be strengthened by further evidence to indicate that the observed effect are not due to extra-exosomal functions and by providing evidence of the impact of Prp40 mutations on neuronal and glial cell phenotypes.

Specific Comments

- The paper should include ‘cellular level’ analyses to look at the phenotypes of neuronal or glial cell populations across developmental times points, e.g. in embryo or larval specimens. Do specific neuronal or glial cell populations display morphological abnormalities that would be consistent with altered expression of some of the transcripts that are found to be upregulated by RNA-seq?

- Could the authors explain their interpretation of why glial cell depletion would severely impact fly eclosion, with little effect on lifespan?

- Is there a reason why G11A/Df(2L) flies were not included in the analysis presented in Fig.2C?

- One question that emerges when considering RNA exosome components is whether there is any evidence for extra-exosomal functions that could potentially explain some of the phenotypes observed in the study? The authors should comment on this and, if feasible, provide some data to address what proportion of Rpr40 (either wt or mutant forms) are associated with other components of the exosome (e.g. gel filtration chromatography, co-IPs…).

- There seem to be some inconsistencies with the numbers of transcripts listed between the various diagrams in Fig.6, i.e. the number o upregulated transcripts don’t match between panels B-C and D-E. The authors should double check these numbers and clarify things in the results section if necessary.

- Typos:

o p.2, files should be flies

o p.8, nulls alleles should be null alleles.

**Have all data underlying the figures and results presented in the manuscript been provided?**

Reviewer #1: No: raw RNAseq datasets should be made available (FASTQ or similar).

Reviewer #2: Yes

PLOS authors have the option to publish the peer review history of their article (what does this mean?). If published, this will include your full peer review and any attached files.

Reviewer #1: No

Reviewer #2: No

---

## [Decision Letter · Decision Letter 1]

1 Jun 2020

Dear Dr. Corbett,

We are pleased to inform you that your manuscript entitled "A Drosophila Model of Pontocerebellar Hypoplasia Reveals a Critical Role for the RNA Exosome in Neurons" has been editorially accepted for publication in PLOS Genetics. Congratulations!

Please address the minor issues mentioned by Reviewer #2 (see below) as you prepare your final draft for the production team (the editorial team will not need to re-evaluate).

You will also need to attend to some formatting changes, which you will receive in a follow up email. Please be aware that it may take several days for you to receive this email; during this time no action is required by you. Please note: the accept date on your published article will reflect the date of this provisional accept, but your manuscript will not be scheduled for publication until the required changes have been made.

Yours sincerely,

Hugo J. Bellen, Ph.D., D.V.M.

Guest Editor

PLOS Genetics

Gregory P. Copenhaver

Editor-in-Chief

PLOS Genetics

Comments from the reviewers (if applicable):

Reviewer's Responses to Questions

**Comments to the Authors:**

Reviewer #1: The revised manuscript by Morton et al. is substantially improved and addresses many of my prior concerns. The new data highlighting mushroom body maldevelopment enhances the relevance of the Drosophila model. In the future, it would be helpful if the revised manuscript included highlighting or tracked changes keyed to the response document to facilitate ease of review. Will the RNA-seq data be deposited for public access? I did not find a clear statement to this effect, but I may have missed this.

Reviewer #2: Overall, the authors have been overall very receptive to reviewer comments and have addressed most of the concerns raised by performing additional experiments. Nevertheless, the following points should be addressed:

Major comments :

Is there a reason why D87A/Df(2L) flies were not included in Fig 5? It would be interesting to complete the figure by adding the D87A/Df(2L) model. This would help explain and reinforce the differences of the lifespan between G11A/G11A and D87A/Df(2L) models, which have the same hatching percentage. This data should be discussed

The data presented in Fig 6A and Fig 2B seem contradictory. Indeed, rescue control flies have a higher hatching percentage compared to neuronal rescue flies (Fig 6A), but the neuronal depletion of Rrp40 has an important impact on hatching. Authors should discuss why the neuronal rescue does not lead to higher hatching frequency.

Minor comments :

Page 8 : “ In contrast, Rrp40wt control flies eclose at a rate comparable to wildtype.”

These data are not available in the figure 3B

Page 11 : « S6A-D Fig » should be « S5A-D Fig ».

Page 12 : « S7 » Fig should be « S6» Fig.

The number of ncRNA in Fig 8E for D87A/D87A does not match with the Fig 8F (124 vs 125).

Verify if the numbers of transcripts in the text (page 11) to describe Fig 8B,C,E, F match with the supplementary tables 2 and 3.

Supplementary data 6 (S6 fig) is exactly the same figure as 9B-D, but the scale is different. Supplementary data 6 is not relevant.

**Have all data underlying the figures and results presented in the manuscript been provided?**

Reviewer #1: None

Reviewer #2: Yes

PLOS authors have the option to publish the peer review history of their article (what does this mean?). If published, this will include your full peer review and any attached files.

Reviewer #1: No

Reviewer #2: No

**Data Deposition**

http://datadryad.org/submit?journalID=pgenetics&manu=PGENETICS-D-19-01373R1

**Press Queries**

---

## [Editor Report · Acceptance letter]

1 Jul 2020

PGENETICS-D-19-01373R1 

A Drosophila Model of Pontocerebellar Hypoplasia Reveals a Critical Role for the RNA Exosome in Neurons 

Dear Dr Corbett, 

We are pleased to inform you that your manuscript entitled "A Drosophila Model of Pontocerebellar Hypoplasia Reveals a Critical Role for the RNA Exosome in Neurons" has been formally accepted for publication in PLOS Genetics! Your manuscript is now with our production department and you will be notified of the publication date in due course.

With kind regards,

Jason Norris

PLOS Genetics

On behalf of:
